# SUMO-targeted Ubiquitin Ligases as crucial mediators of protein homeostasis in *Candida glabrata*

**Dipika Gupta, Renu Shukla, Krishnaveni Mishra** *

Department of Biochemistry, School of Life Sciences, University of Hyderabad, Hyderabad, India

* krishnaveni@uohyd.ac.in, kmsl.uoh@nic.in

## Abstract

*Candida glabrata* is an opportunistic human pathogen, capable of causing severe systemic infections that are often resistant to standard antifungal treatments. To understand the importance of protein SUMOylation in the physiology and pathogenesis of *C. glabrata*, we earlier identified the components of SUMOylation pathway and demonstrated that the deSUMOylase CgUlp2 is essential for pathogenesis. In this work we show that the CgUlp2 is essential to maintain protein homeostasis via the SUMO-targeted ubiquitin ligase pathway. The dual loss of deSUMOylase and specific ubiquitin ligase, CgSlx8, results in heightened protein degradation, rendering the cells vulnerable to various stressors. This degradation affects crucial processes such as purine biosynthesis and compromises mitochondrial function in the mutants. Importantly, the absence of these ubiquitin ligases impedes the proliferation of *C. glabrata* in macrophages. These findings underscore the significance of SUMOylation and SUMO-mediated protein homeostasis as pivotal regulators of *C. glabrata* physiology and capacity to survive in host cells. Understanding these mechanisms could pave the way for the development of effective antifungal treatments.

## Author summary

Opportunistic fungal infections are a major cause of mortality across the world. There are a limited number of antifungal drugs and many fungi are resistant to one or more of these. Therefore, there is an unmet need for new antifungals. As fungi are evolutionarily close to humans, identifying targets that are unique or sufficiently different from human counterparts is crucial. Our laboratory has been investigating if the post-translational modification pathway, SUMOylation, can be an effective target. To this end, we are studying the importance of this pathway in the pathogenic yeast, *Candida glabrata*, which is the second most common causative agent for candidiasis. We had earlier identified the components of this pathway and showed that this reversible protein modification was important for pathogenesis. Here, we investigated the mechanistic basis of the role of SUMOylation-deSUMOylation in *C. glabrata* infection and find that the balanced activity of this pathway is critical to maintain protein homeostasis in the cell. Perturbation of this pathway leads to unwarranted protein degradation rendering the cell incapable of

**Data Availability Statement:** All DATA is in the manuscript or Supporting information.

**Funding:** KM acknowledges support from SERB (EMR/2017/003020) for this project. DG thanks CSIR for fellowship and RS was supported by DST-

WOS-A-SR/LS/464/2018. KM lab is supported by funds from DBT, SERB and Institution of Eminence, UoH (RC3-21-060). The funders had no role in study design, data collection and analysis, decision to publish, or preparation of the manuscript.

combating stress including survival in host cells. Therefore, the SUMO-mediated protein homeostasis pathway in *C. glabrata* could be targeted for new antifungal development.

## Introduction

Fungal infections are a global health problem. Estimates say over 150 million severe cases of fungal infections occur worldwide, resulting in approximately 1.7 million deaths per year world-wide due to lack of effective antifungals [1]. *Candida glabrata* is listed as a high-priority pathogen by the WHO in the Fungal priority pathogen list (FFPL) as one of the key causative agents of candidiasis and due to its innate resistance to high levels of antifungals [2–7]. The increased antifungal resistance causes high level of mortality in immunocompromised individuals [7]. Therefore, there is an unmet need for new and effective antifungals. A key problem in antifungal research is the close evolutionary relatedness between animals and fungi, therefore, identifying effective targets that are not toxic to humans is challenging. One of the potential targets could be post-translational modifications, because they regulate gene expression in fungi under stress including infection of host cells [8]. Ubiquitin and SUMO (Small Ubiquitin-like Modifier) belong to the Ubiquitin-like modifier proteins (Ubls) that are conjugated to lysine residues of the target protein to regulate their function and localization. They are also found to be actively involved in multiple cellular activities like cell stress response and overall maintenance of cellular homeostasis [9,10].

The coupling of SUMO and Ubiquitin to their substrate is a multi-step process. In SUMOylation, the translated SUMO protein is proteolytically processed at the C-terminus to reveal a di-glycine motif in the mature SUMO. The C-terminal glycine is conjugated to the epsilon amino group of lysine in target proteins. First, in an ATP-dependent reaction, the mature SUMO moiety are attached to E1 activating enzyme (Aos1 and Uba2 in *S. cerevisiae*) and then transferred to E2 conjugating enzyme (Ubc9) through a transesterification reaction [11,12]. E2 finally binds to E3 ligases (Siz1, Siz2, Mms21, Zip3) to facilitate the transfer of SUMO moieties to the lysine residues of the target substrate [9,10,13–15]. SUMO can be removed from targets by the deSUMOylases, Ulp1 and Ulp2, thereby ensuring the reversibility of SUMO modification and also maintain a pool of free SUMO in the cell for the SUMOylation cycle [16–18]. Ulp1 is the enzyme that also catalyses the maturation of SUMO. Both Ulp1 and Ulp2 have several targets that they deSUMOylate and specificity is partly imposed by the differential localization of the two deSUMOylases. Ulp1 is associated with the nuclear pore complex while Ulp2 is in the nucleoplasm in *S. cerevisiae*. While yeast has two deSUMOylases, there are six known deSUMOylases of the SENP class in mammals. SENP-1, SENP-2, SENP-3, SENP-5 are evolutionarily closer to Ulp1 while SENP-6, and SENP-7 are more similar to Ulp2 [18].

Protein SUMOylation is conserved across eukaryotes. Monomer of SUMO can be attached to a single lysine residue of a target or to multiple lysine residues of a target protein generating either monoSUMOylated or multiSUMOylated proteins. In addition, polySU-MOylation is a special form of SUMOylation where polymeric chains of SUMO are produced when another SUMO molecule is added to the internal lysine residues of the SUMO present on target proteins. In yeast, the E3 ligase Siz2 is thought to be the primary ligase that produces polySUMO and Ulp2 is the deSUMOylase that effectively removes polySUMO chains [19]. PolySUMO serves as a platform for the recruitment of SUMO-targeted ubiquitin ligases (STUbLs) that recognize polySUMO chains. STUbLs are E3-ubiquitin ligases, and in yeast, the well-studied STUbLs are the RING domain containing Slx5/Slx8 heterodimer and Uls1 [20]. In mammals, to date, two STUbLs have been identified, namely, RNF4 and

RNF11. Thus the deSUMOylases, especially Ulp2, play a key role in balancing SUMOylation and ubiquitination of target proteins and as STUbL mediated ubiquitination usually leads to protein degradation, Ulp2 is critical to maintain protein homeostasis in several complexes including chromatin structure, kinetochore assembly, DNA repair in *S. cerevisiae* [21–24].

In our laboratory we have shown that protein SUMOylation is essential for *Candida glabrata* and the pathway is critical for pathogenesis [25]. Other laboratories have shown that SUMOylation plays a major role in pathogenesis of fungi like *Candida albicans*, *Aspergillus sp.* and *Magnaportha sp* [26–29]. After demonstrating that loss of SUMO pathway genes leads to reduced pathogenicity of *C. glabrata*, we also identified the genes involved in SUMOylation and the SUMO-mediated protein homeostasis pathway in *C. glabrata* and other fungi [30]. In this manuscript, we have characterized the SUMO-regulated protein homeostasis pathway of *C. glabrata*. We demonstrate that imbalance in STUbLs and the deSUMOylase leads to increased protein degradation, reducing the capability of *C. glabrata* to combat stress. We also show that several pathways, particularly purine biosynthesis, mitochondrial metabolism and protein homeostasis pathways are affected. Overall this results in reduced fitness and reduced ability of *C. glabrata* to infect and proliferate in host cells, thus providing a promising pathway that can be targeted for antifungal development.

## Results

### STUbLs are required for key cellular processes

In our previous work, we had identified the homologues of the SUMOylation and the SUMO-targeted ubiquitin ligase (STUbL) pathways in fungi [30]. We found that all fungi had at least one of the STUbLs suggesting that the SUMO-dependent ubiquitin pathway of protein homeostasis could be important for the physiology of fungi [30]. To understand the importance of STUbLs in *C. glabrata*, we used BLAST analysis to compare homologies between the *S. cerevisiae* and *C. glabrata* proteins. These proteins include Slx5, Slx8, and Uls1 and S1 Table displays their percent similarity across the complete protein sequence. Uls1 is the most conserved of them, having 60% similarity between *C. glabrata* and *S. cerevisiae* (S1 Table). Both the Slx5 and Slx8 share over 45% protein sequence similarity between *C. glabrata* and *S. cerevisiae*. Additionally, STUbLs have conserved regions of multiple SIM motifs and a RING motif (S1 Fig). SIM motifs are SUMO interacting motifs that bind to SUMO moiety while the RING domain is required for catalysis and protein-protein interaction [31]. CgUls1, like the *S. cerevisiae* counterpart, has a "SNF2-like translocase" domain (S1 Fig).

We generated STUbL-deficient strains of *C. glabrata* to study the impact of STUbL disruption on the physiology and infectivity of the organism. Using a fusion PCR and homologous recombination-based method, we were able to create deletion strains for the *CgSLX5*, *CgSLX8*, and *CgULS1* genes. In order to better understand how STUbLs interact among themselves and with SUMO protease *CgULP2* in the pathobiology of C. glabrata, we also created double deletion strains of STUbLs and STUbLs with *CgULP2*. We could not generate *Cgslx8ΔCguls1Δ* double mutant despite several attempts, possibly because loss of both the STUbLs is lethal in *C. glabrata*.

We first performed the phenotypic characterization of the STUbL mutants in *C. glabrata* to study the role of SUMO-targeted ubiquitin ligases in cell physiology. Growth analysis of single deletion strains of *Cgulp2Δ*, Cg*slx5Δ*, Cg*slx8Δ*, and Cg*uls1Δ*, as well as double deletion strains of *Cgslx5ΔCgslx8Δ*, *Cgslx5ΔCguls1Δ*, *Cgulp2ΔCgslx8Δ*, and *Cgulp2ΔCguls1Δ*, showed that the mutant strains of *Cgulp2Δ* grew slower than the wild-type (WT) as expected [25]. In addition, *Cgulp2ΔCgslx8Δ* double mutant grew much slower compared to WT and the *Cgulp2Δ* in both YPD and RPMI even at conducive temperatures (Figs 1A and 2B).

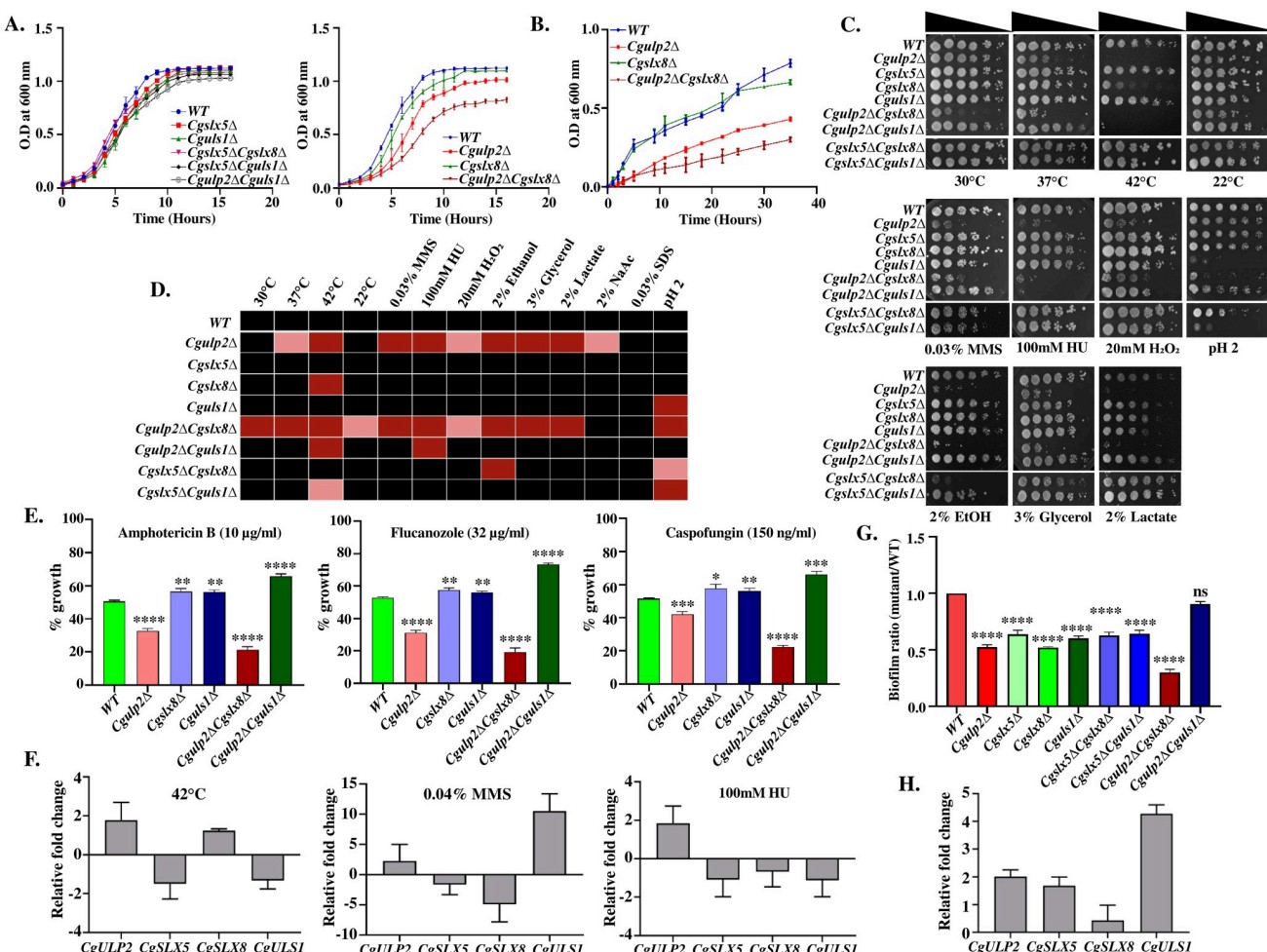

**Fig 1. STUbL genes are crucial for the maintenance of cellular homeostasis.** Overnight grown cultures of indicated *C. glabrata* strains were inoculated in **(A)** YPD and **(B)** RPMI complete medium to an initial $A_{600}$ of 0.1. Absorbance at 600nm was measured at regular time intervals. Data represent the mean of biological triplicates. **C.** Sensitivity of the STUbL deletion strains to various stress conditions. Two microlitres of ten-fold serial dilutions of exponentially growing cultures were spotted on indicated plates and images were taken after 2–4 days of incubation. **D.** Heat map illustrating growth of STUbL mutants in the presence of diverse stress-causing agents. The sensitivity of STUbL mutants of *C. glabrata* to various treatments is represented as a heat map with black color (not sensitive; full growth) and different shades of red color (dark red indicates a high level of sensitivity). **E.** Bar graph depicting the survival of wild type, *Cgulp2Δ*, *Cgslx8Δ*, *Cguls1Δ*, *Cgulp2ΔCgslx8Δ*, and *Cgulp2ΔCguls1Δ* mutants treated with 32 µg/ml fluconazole, 150 ng/ml caspofungin, and 10 µg/ml amphotericin B. Survival was calculated relative to untreated control samples. Data represent the average of three independent experiments, with error bars indicating standard deviation (SD). Statistical significance was assessed using a two-tailed student's t-test (*- $p \leq 0.01$; **- $p \leq 0.005$; ***- $p \leq 0.0005$; ****- $p \leq 0.0001$). **F.** Expression studies of SUMO protease and STUbL genes under various stress conditions (42˚C, 0.04% MMS, 100mM HU) by performing RT-PCR. Data was normalized with untreated wild type cell and the graphs show mean and SD of biological triplicates. **G.** Biofilm formation was recorded for *C. glabrata* mutants vs wild type using XTT assay. Data represent the mean and SD of three independent experiments. Significance was assessed by two-tailed unpaired Student's t-test with ****- $p \leq 0.0001$; ns- non-significant. **H.** Biofilm from the wild type cells were used for RNA isolation and cDNA conversion. RT-PCR was then performed to check the expression level of SUMO protease *CgULP2* and STUbL genes. The fold change for each gene in the biofilm-forming wild type cells was calculated with respect to planktonic cells. The experiment was done in biological triplicate and mean was plotted. The error bar shows the standard deviation of the mean.

Earlier studies have shown that *Cgulp2Δ* mutant grew slowly at 42˚C and was vulnerable to the DNA-alkylating agent methyl methanesulfonate (MMS), replication fork staller hydroxy-urea (HU), and oxidative stress-inducing agent hydrogen peroxide ($H_2O_2$) [25]. Single mutant of the STUbLs grew similar to wild type under most conditions, the exceptions being mild temperature sensitivity of *Cgslx8Δ* at 42˚C and slow growth of *Cguls1Δ* in low pH media

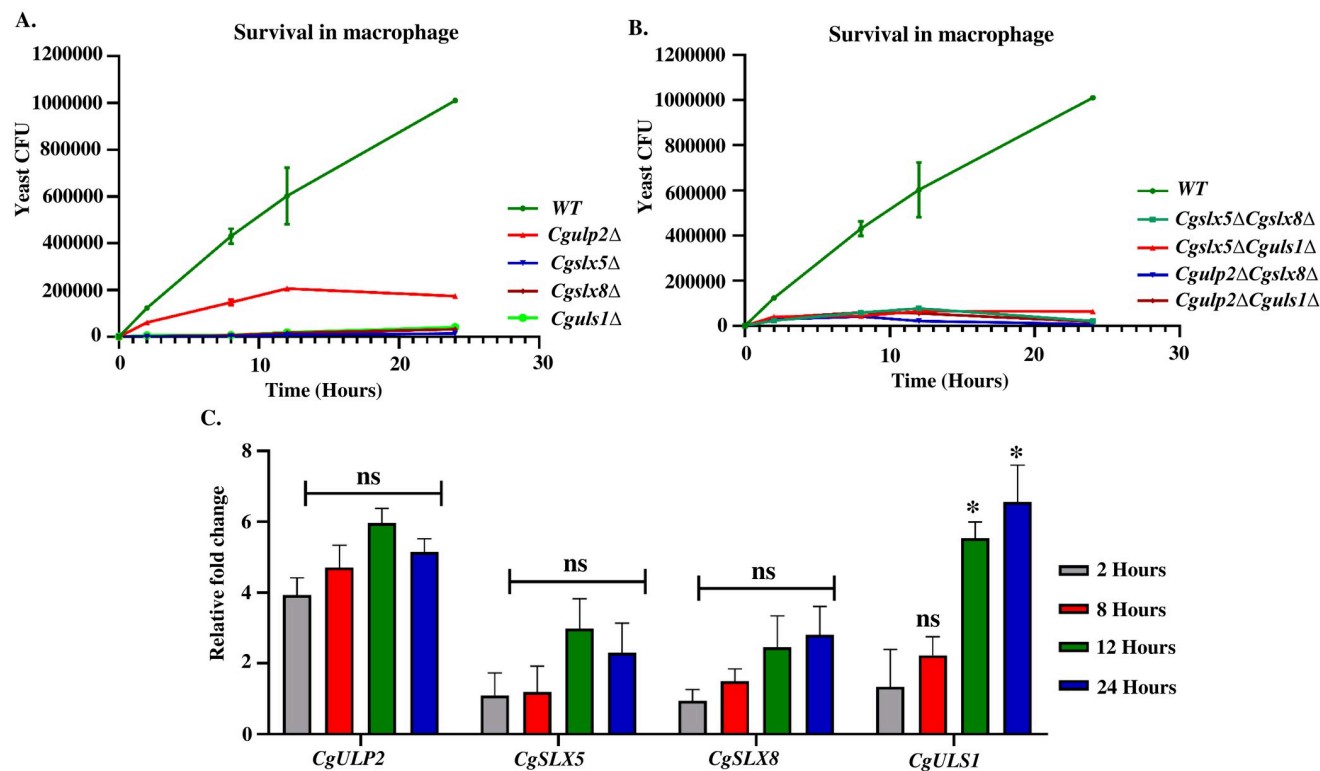

**Fig 2. Reduced survival and replication inside macrophage by STUbL mutants of *C. glabrata*. A.** and **B.** $2.5 \times 10^6$ cells of the indicated *C. glabrata* strains were added to phorbol 12-myristate 13-acetate-differentiated THP-1 cells in a 24-well plate, and non-phagocytosed *C. glabrata* cells were removed after 2 hours by washing. Intracellular yeast cells were recovered at the indicated time intervals by lysing THP-1 cells in water and plating appropriate dilutions of lysates on the YPD agar. After 24–48 hours of incubation at 30˚C, yeast colonies were counted. Mean of three independent experiments are shown alongwith SD. **C.** Expression levels of STUbL genes of the macrophage internalized *C. glabrata* cells. Mean of three independent experiments are shown alongwith SD (*- $p \leq 0.01$; ns- non-significant, two-tailed unpaired Student's t-test).

(Figs 1C, 1D and S2). Contrary to *Cgslx8Δ* and *Cguls1Δ*, the growth of the *Cgslx5Δ* mutant remains unaffected under diverse stress conditions (Figs 1C, 1D and S2). Both *Cgslx5ΔCguls1Δ* and *Cgulp2ΔCguls1Δ* double mutants were temperature sensitive for growth at 42˚C when compared with WT (Figs 1C, 1D and S2). Interestingly, *Cgulp2ΔCguls1Δ* double mutants suppressed the sensitivity of *Cgulp2Δ* single mutant to oxidative stress (20 mM $H_2O_2$), DNA damage (0.03% MMS) and low growth on non-fermentable carbon sources (2% ethanol, 3% glycerol, 2% lactate, and 2% sodium acetate). *S. cerevisiae* strains with *slx5Δ* and *slx8Δ* deletion are sensitive to cell membrane stress (0.03% SDS) [32]. However, neither the single nor the double mutants of *C. glabrata* were sensitive to SDS (Fig 1D).

The *Cgulp2ΔCgslx8Δ* strain was generally more sick and grew poorly at 30˚C compared to either single mutants and it grew better at low temperature (22˚C) than at 30˚C (Figs 1C, 1D and S2). *Cguls1Δ*, *Cgslx5ΔCgslx8Δ*, *Cgslx5ΔCguls1Δ*, and *Cgulp2ΔCgslx8Δ* were sensitive to acidic pH as well (Fig 1D). However, *Cgulp2ΔCgslx8Δ* was more sensitive to all stress conditions compared to either single mutants i.e., *Cgslx8Δ* exacerberated the sensitvity of *Cgulp2Δ* single mutant to stress. The phenotypes of increased stress sensitivity in single and double deletion strains of SUMO-targeted ubiquitin ligases in various stress conditions were rescued by ectopic expression of *CgULP2* and STUbL genes, indicating, indeed, the phenotypes observed are due to loss of STUbLs (S3 Fig). In summary, these data show that the loss of *Cgslx5* has minimal effect on the various conditions tested here; *Cgslx8Δ* exacerbates multiple

phenotypes of *Cgulp2Δ* and *Cguls1Δ* suppresses sensitivity of *Cgulp2Δ* to several stress conditions.

As the STUbLs appeared to contribute to stress resistance in *C. glabrata*, we also assessed antifungal susceptibility of these mutants. We first established that 50% growth inhibition of wild type *C. glabrata* was at 32 μg/ml fluconazole, 150 ng/ml caspofungin, and 10 μg/ml amphotericin B (Fig 1E). The *Cgulp2Δ* mutant showed heightened sensitivity to these antifungal agents, with survival of 31%, 42%, and 32%, respectively, indicating a reduced tolerance. In contrast, the *Cgslx8Δ* and *Cguls1Δ* mutants exhibited slightly higher resistance, maintaining survival between 56–58% across the treatments. However, the *Cgulp2ΔCgslx8Δ* double mutant showed the greatest sensitivity, with only 19%, 22%, and 21% survival under the same conditions, highlighting an increase in vulnerability when both the genes were deleted. Interestingly, the *Cgulp2ΔCguls1Δ* double mutant displayed a more robust antifungal tolerance, with survival of 73%, 66%, and 65%, suggesting that the loss of *Cguls1* partially mitigates the stress experienced by the *Cgulp2Δ* single mutant (Fig 1E).

As the loss of STUbLs had differential sensitivities to various stress conditons tested here, it is possible that they are specifically required to combat these conditons. Therefore we analysed the expression of the SUMO protease *CgULP2* and STUbL genes under thermal stress (42°C), 0.04% MMS and hydroxyurea treatment by quantitative RT-PCR (Fig 1F). The phenotypes observed largely correlated with the expression analyses of the SUMO protease *CgULP2* and STUbL genes under thermal stress (42°C), MMS and hydroxyurea treatment. For instance, in the aforementioned stress conditions, the lack of *CgULP2* caused significant growth anomalies and we see elevated expression of *CgULP2* when exposed to these stress. However, *CgSLX5*, *CgSLX8*, and *CgULS1* genes were downregulated in response to hydroxyurea treatment; and these strains do not display any sensitivity to hydroxyurea. *Cgslx8Δ* was sensitive to higher temperatures, and its expression increases when subjected to thermal stress while *CgSLX5* and *CgULS1* levels were lower. When treated with MMS, the *CgULS1* gene upregulation was seen, whereas *CgSLX5*, and *CgSLX8* were downregulated (Fig 1F). These findings collectively suggest that specific SUMO-targeted ubiquitin ligases are essential for resistance to heat and DNA damage.

Biofilm formation by *C. glabrata* on the biotic (body tissue) and abiotic (plastic and medical devices) surfaces are major virulence factors [33]. We therefore tested the ability of these mutants to form biofilm and compared them with wild type strains. Interestingly, all single mutants of STUbL had reduced capacity of form biofilms and the double mutants of *Cgulp2ΔCgslx8Δ* were more compromised than either single mutant (Fig 1G). *Cgulp2ΔCguls1Δ*, on the other hand, formed biofilm similar to that of WT. This follows the same trend that we have observed for stress conditions tested in that the double mutant of *Cgulp2ΔCguls1Δ* suppresses the biofilm formation defect of both the single mutants, *Cgulp2Δ* and *Cguls1Δ*. Further, to assess if these phenotypes indicated the requirement of these gene products for biofilm formation, we checked the expression of deSUMOylase *CgULP2* and STUbL (*CgSLX5*, *CgSLX8*, and *CgULS1*) genes. *CgULS1* was found to be highly upregulated which was followed by *CgULP2* and *CgSLX5*, while *CgSLX8* levels were similar to wild type (Fig 1H). Interestingly, while *CgULS1* and *CgULP2* expression is upregulated in biofilms, the double mutant *Cgulp2ΔCguls1Δ* is competent for biofilm formation, unlike the two single mutants. This indicates that there is a connection between Ulp2 substrates and Uls1-mediated protein degradation that leads to different outcomes for the single and double mutants.

## STUbLs are essential for the survival of *C. glabrata* within the macrophage

We studied the ability of STUbL mutants to survive and proliferate in macrophages using the colony-forming unit (CFU) assay. For this, cultured human THP-1 macrophages were infected

with equal numbers of wild type, *Cgulp2Δ*, and STUbL mutants. Internalized *C. glabrata* were then extracted from the macrophages at the indicated time points and plated on YPD solid medium to obtain the live cell count. Based on our previous studies, we know that after 8 hours of addition of *C. glabrata* to macrophages, the internalized *C. glabrata* cells start to multiply and this was captured in the WT infections in this experiment as well. However, we observed that the survival and replication of all the internalized STUbL mutants of *C. glabrata* were much less than the wild type (Fig 2A and 2B). Also, while growth in YPD was similar in all strains except *Cgulp2Δ* and the *Cgulp2ΔCgslx8Δ* as described above, growth in RPMI medium was 6–8 fold lower in *Cgulp2Δ* and all STUbL mutants as compared to the wild type (S4 Fig) suggesting that STUbLs are required for optimal growth in alternate culture media.

To test if deSUMOylase and SUMO-targeted ubiquitin ligases were induced upon infection in macrophages, we checked the expression of deSUMOylase *CgULP2* along with STUbL genes *CgSLX5*, *CgSLX8*, and *CgULS1* at different time points. Fig 2C shows the changes in expression during the time course of infection. The expression of *CgULP2* was upregulated as early as 2 hours of infection, which remained high upto 24 hours. Importantly, expression of *CgSLX5*, *CgSLX8*, and *CgULS1* was also found to be induced; while the *CgSLX5* and *CgSLX8* transcripts show a modest increase, *CgULS1* had higher levels of expression (Fig 2C). Notably, these findings also corroborate with the CFU assay, which revealed that single mutants of *Cgulp2Δ*, *Cgslx5Δ*, *Cgslx8Δ*, and *Cguls1Δ* had reduced intracellular proliferation and had a low CFU count (Fig 2A and 2B). This supports the notion that *CgULP2*, *CgSLX5*, *CgSLX8*, and *CgULS1* are required for the survival and proliferation of pathogen in the host.

## Protein SUMOylation in STUbL mutants of *C. glabrata*

Loss of function of Ulp2 was earlier shown to reduce the amount of SUMOylated proteins [25]. We had reasoned that accumulation of polySUMOylated proteins led to increased degradation of proteins via the SUMO-targeted Ubiquitin ligases, resulting in lower levels of SUMOylated protein in *Cgulp2Δ* strains. Therefore, we hypothesized that double deletion strain of both *Cgulp2* and STUbLs will lead to the retention of higher levels of polySUMOylated proteins. In order to determine the SUMO proteome in each of the STUbL mutants, we used an anti-FLAG antibody to perform western blots on whole cell lysates from the *C. glabrata* STUbL mutants augmented with dual tagged CgSmt3 protein (carrying His6 and FLAG epitopes at the N-terminus). The lack of higher molecular weight SUMOylated proteins in the *Cgulp2* mutant was confirmed (Fig 3A). In the case of STUbL single mutants, only *Cgslx5Δ* was found to have higher molecular weight SUMOylated proteins (Fig 3A). In principle, STUbL deletion strains should accumulate higher molecular weight polySUMOylated proteins [34]. It is possible that the absence of higher molecular weight SUMO conjugates in the *Cgslx8Δ* and *Cguls1Δ* single mutants suggests that in the absence of one of the STUbLs, another might compensate and one STUbL gene may be sufficient to maintain homeostasis. In this context, it is interesting to note that the *Cgslx8ΔCguls1Δ* double mutant could not be generated, suggesting at least one of these STUbLs was essential for survival. Intriguingly, accumulation of higher molecular weight SUMOylated proteins was seen in the double mutants of *Cgulp2ΔCguls1Δ*, and *Cgslx5ΔCguls1Δ*. This suggests that Uls1 could be the primary ubiquitin-ligase for many of the SUMOylated proteins and in the absence of Uls1, these proteins are stabilized (Fig 3A, Lane 7 and 9). Of note, we have demonstrated that *Cgulp2ΔCguls1Δ* double mutant was able to restore *Cgulp2Δ* mutant growth under a variety of stress conditions (Fig 1C and 1D) and interpret the appearance of these SUMOylated proteins as a reason for the restoration of growth in the double mutant. Importantly, the *Cgulp2ΔCgslx8Δ*, which had exacerbated the *Cgulp2Δ* phenotypes, not only grew very slowly, it also has the least amount of

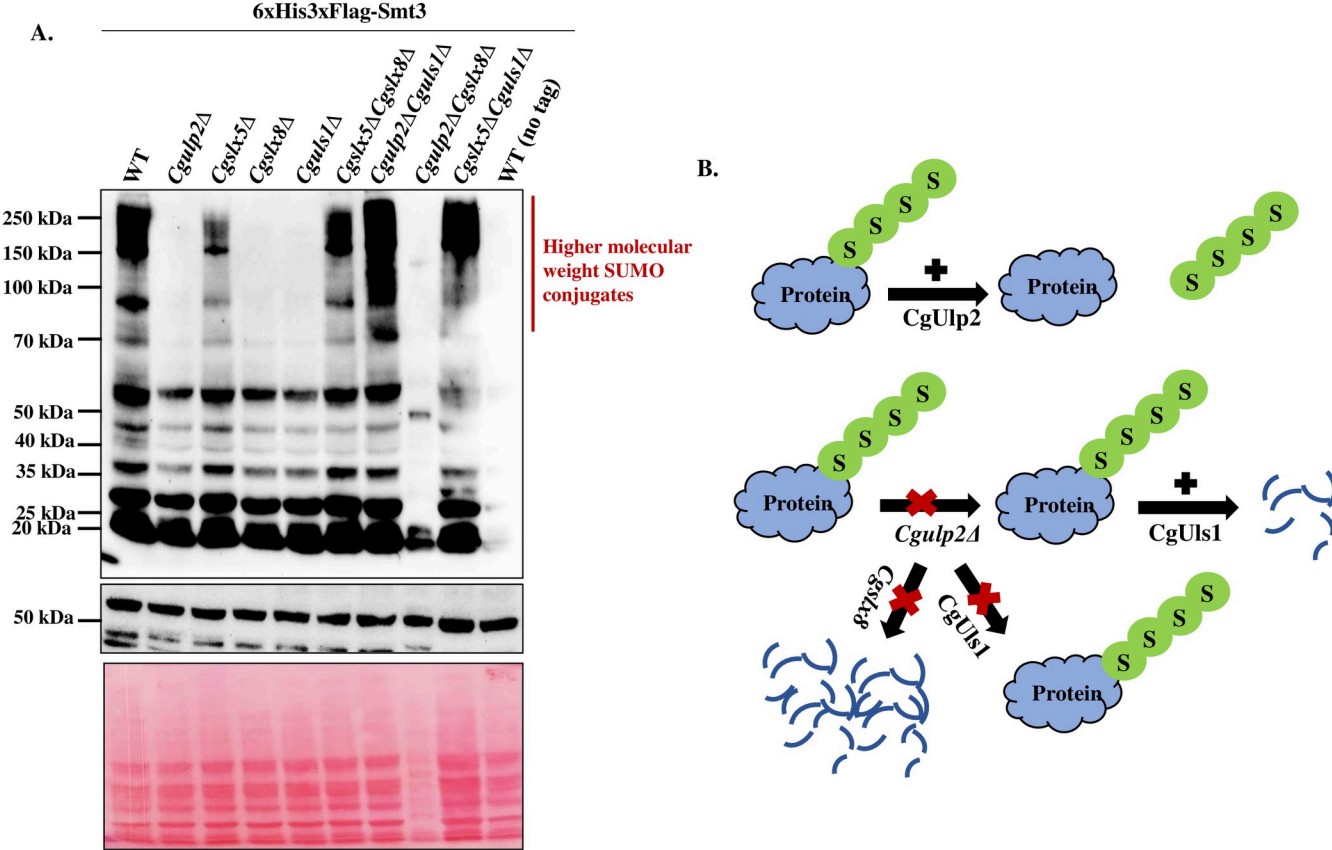

**Fig 3. Differential SUMOylation pattern in STUbL mutants of *C. glabrata*. A.** Strains were grown in YPD agar and an equal number of cells were taken for whole-cell extracts. Tubulin was used as a loading control. The blot was stained with Ponceau S to show the level of protein degradation in *Cgulp2ΔCgslx8Δ* double mutant in comparison to WT and other mutants. **B.** The diagram illustrates the increased protein degradation in *Cgulp2ΔCgslx8Δ* and the restoration of protein levels in the *Cgulp2ΔCguls1Δ* double mutant.

SUMOylated proteins, suggesting enhanced degradation of proteins (Fig 3A-Lane 8 and Fig 3B).

Each of these enzymes affects growth phenotypes and SUMOylated products differently. Based on these observations, we reason that because the polySUMOylated proteins are not degraded when *Cguls1* is lost in the *Cgulp2Δ* mutants, the *Cgulp2Δ* reduced growth phenotype is suppressed in the double mutant. On the other hand, *Cgulp2Cgslx8Δ* cells lose more protein as seen in the western blot, and get sicker (Fig 3A). Based on this and the observation that single mutants of STUbLs do not accumulate higher molecular weight SUMOylated proteins, we hypothesize that *Cgslx8* deletion possibly upregulates Uls1 activity and leads to increased protein degradation making the *Cgulp2Cgslx8Δ* particularly vulnerable. An alternative explanation is that CgUls1 is the main STUbL, however, several essential proteins required for strong growth may be degraded in *Cgslx8Δ* mutant or accumulated as polySUMOylated proteins and induce toxicity leading to compromised growth.

## CgUls1 protein is upregulated in STUbL mutants

To test whether CgUls1 is upregulated in *Cgslx8* deletion, we created a construct of dual-tagged 6xHIS-3xFLAG-*CgULS1* (HF-*CgULS1*) under its own promoter and confirmed the expression of CgUls1 protein in the *Cguls1Δ* mutant by western blot using an α-flag antibody. *Cguls1Δ*

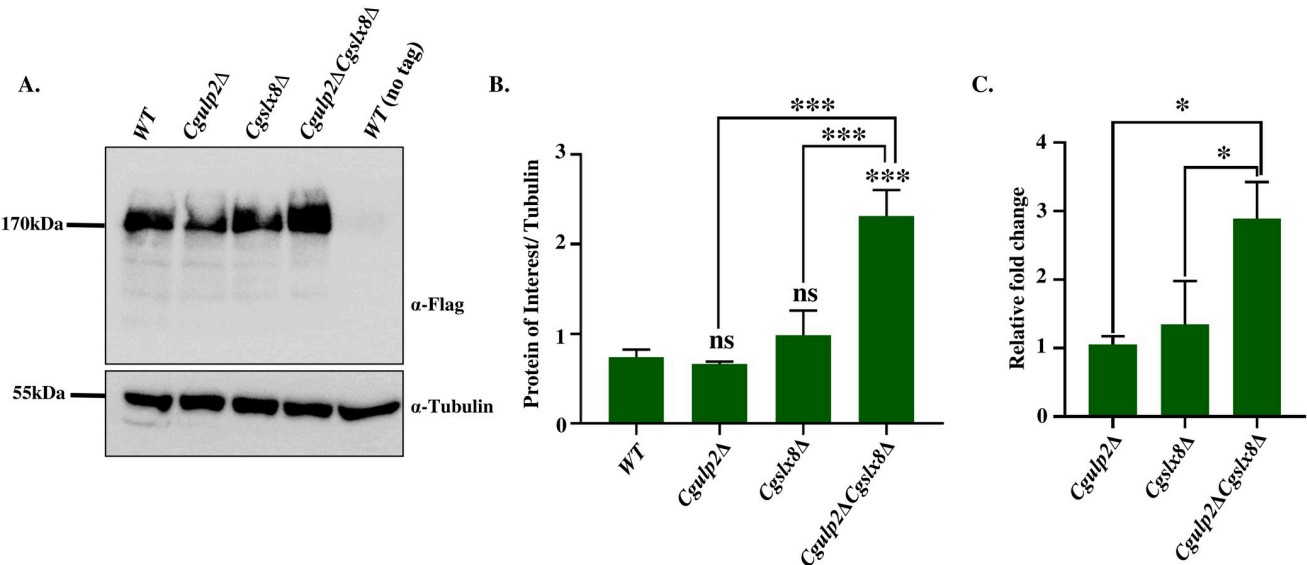

**Fig 4. Upregulation of CgUls1 protein in STUbL mutant. A.** Western blot analysis to show the expression level of CgUls1 protein in STUbL mutants using anti-FLAG antibody. Tubulin was used as a loading control. **B.** Quantification of the western blot. ***- p ≤ 0.0005; ns- non-significant; one-way ANOVA. **C.** YPD grown wild type, *Cgulp2Δ*, *Cgslx8Δ*, and *Cgulp2ΔCgslx8Δ* cells were used for RNA isolation and cDNA conversion. RT-PCR was then performed to check the expression level of *CgULS1* gene in these mutants. Data represents the mean and standard deviation of three independent experiments. Statistical significance was calculated using one-way ANOVA (*, p < 0.05).

was sensitive to DNA-damaging agents at higher MMS (0.04%) concentration. We tested the complementation of HF-*CgULS1* in the *Cguls1Δ* mutant in this condition and we found that HF-*CgULS1* restored the phenotype of the *Cguls1Δ* mutant (S5A Fig).

We checked the level of CgUls1 protein in WT, *Cgulp2Δ*, *Cgslx8Δ*, and *Cgulp2ΔCgslx8Δ* deletion strain and found higher amounts of CgUls1 protein in *Cgulp2ΔCgslx8Δ* double deletion mutant in comparison with wild type and *Cgulp2Δ*, *Cgslx8Δ* single mutants (Fig 4A and 4B), suggesting that there was upregulation of CgUls1 in the *Cgulp2ΔCgslx8Δ* mutants. Further, to see if this was transcriptional upregulation, we compared the expression of *CgULS1* gene in these deletion strains by quantitative RT-PCR. Indeed the transcripts were higher in the double mutant compared to *Cgulp2Δ* and *Cgslx8Δ* single mutant (Fig 4C). Of note, we detected higher than wild type levels of CgUls1 protein in *Cgulp2ΔCgslx8Δ* strains in the proteomics data as well (S5B Fig). Together, these findings support our hypothesis that, due to the increased potential activity of the CgUls1 protein in the *Cgulp2ΔCgslx8Δ* mutant, cells lose more protein and are compromised for growth even under conducive conditions.

## Loss of proteins and reduced growth of STUbL mutants is due to enhanced proteasomal degradation

We examined the SUMOylation pattern in *Cgulp2Δ*, *Cgslx8Δ*, and *Cgulp2ΔCgslx8Δ* cells treated with the proteasome inhibitor MG132 to test the possibility that an increase in the accumulation of polySUMOylated proteins in these mutants results in their degradation by the ubiquitin-proteasome pathway. Upon treatment with MG132, degradation of higher molecular weight SUMO-conjugated proteins was prevented, and higher molecular weight SUMOylated proteins could be detected in *Cgulp2Δ* single mutant (Fig 5B, Lane 2) as compared to the untreated mutant strain (Fig 5A, Lane 2). Fewer changes were seen in the *Cgslx8Δ* consistent with the phenotypes seen for this mutant. Most interestingly, in the double mutant of

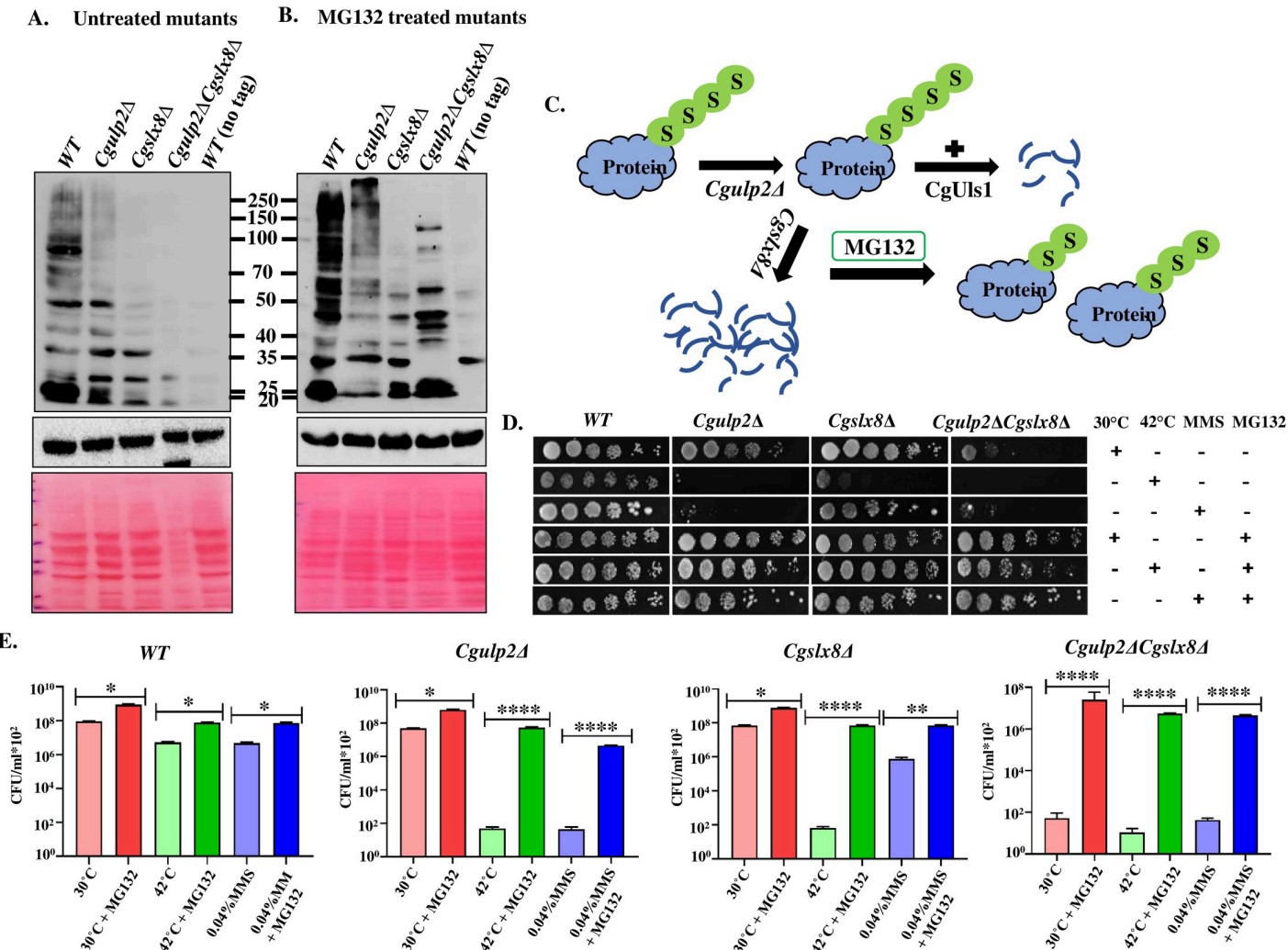

**Fig 5. Loss of proteins and reduced growth of STUbL mutants is due to enhanced proteasomal degradation.** *C. glabrata* strains, both wild type and mutant, were grown overnight in YNB medium at 30°C and 180 rpm with 0.1% proline serving as the only nitrogen source. After cell harvesting, a fresh medium containing 0.003% SDS was used to adjust the $A_{600}$ to 0.5, and the cells were then grown for 4 hours. **A.** Cells without MG132 treatment were used as a control. **B.** Cultures were placed in a 2 ml microcentrifuge tube with 1 ml of YNB-proline media and 200μM MG132, and were then incubated for two hours at 30°C and 180 rpm. Following cell pelleting and whole cell extract preparation, cells were analyzed by western blot using α-Flag antibody. Tubulin was used as a loading control. Blots stained with Ponceau S show protein degradation in *Cgulp2ΔCgslx8Δ* double mutant in untreated cells (A, lane 4) in comparison to WT and recovery in protein levels upon MG132 treatment (B, lane 4). **C.** Diagram shows that MG132 treatment prevents protein degradation in *Cgulp2ΔCgslx8Δ* mutant. **D.** MMS (0.045%) was added to the media and grown for 3 hours after MG132 treatment for 2 hours. Similarly, temperature treatment was given to the cultures for 3 hours. After that, cultures were serially diluted 10-fold, and a 2 μl volume was spotted on the YPD medium. Images of the plates were taken following 2–3 days of incubation at desired temperatures. **E.** Quantification of the growth of MGI32 treated STUbL mutants in various stress conditions. Data represents the mean and SD of at least three independent experiments. Statistical significance was assessed using a two-tailed paired student's t-test (*- p ≤ 0.01; **- p ≤ 0.005; ***- p ≤ 0.0005).

*Cgulp2ΔCgslx8Δ*, with MG132, higher molecular weight SUMO-conjugated proteins were now appearing suggesting prevention of degradation of proteins (Fig 5B-Lane 4 and Fig 5C) compared to the untreated mutant (Fig 5A, Lane 4). Furthermore, the reappearance of higher molecular weight proteins is accompanied by the concomitant restoration in the growth of these mutants (Fig 5D). Also, MG132 treatment could overcome the temperature and MMS sensitivity of the *Cgulp2Δ*, *Cgslx8Δ*, and *Cgulp2ΔCgslx8Δ* mutant (Fig 5D–5E). These findings strongly imply that the reason for slow growth at 30°C and sensitivity to stress inducers in *Cgulp2ΔCgslx8Δ* double mutant was due to excess proteasomal degradation. Together these

results support our hypothesis that loss of Ulp2 leads to the increased accumulation of polySU-MOylated proteins that were then degraded by proteasome via the STUbL pathway leading to deregulated protein homeostasis.

### Identification of global proteome of *Cgulp2Δ*, *Cgslx8Δ*, *Cguls1Δ*, *Cgulp2ΔCgslx8Δ* and *Cgulp2ΔCguls1Δ* deletion strains in comparison with wild type

The results shown in the above sections indicate that knock out of a single STUbL has minimal effect on the growth and capacity of the organism to combat stress when grown in rich media. However, when the deSUMOylase is also inactive, we observe contrasting outcomes. When C*guls1* is deleted in the *Cgulp2Δ*, the strains recover from some of the adverse effects of *Cgulp2Δ*. On the other hand, loss of *Cgslx8Δ* in the *Cgulp2Δ*, leads to exacerbation of pheno-types and rendering the double mutants more compromised for growth in most conditions. We speculate that both the exacerbation and suppression of *Cgulp2Δ* phenotypes are due to differences in the degradation of selected polySUMOylated proteins. In order to obtain an overall idea of the proteome in the absence of these STUbLs and Ulp2, and also to determine if proteins that are part of critical growth pathways are particularly degraded in the STUbL double mutant *Cgulp2ΔCgslx8Δ* and restored in *Cgulp2ΔCguls1Δ*, we conducted tandem mass tag (TMT) labeled quantitative mass spectrophotometry analyses of log-phase grown cells of WT, *Cgulp2Δ*, *Cgslx8Δ*, *Cguls1Δ* single mutants, *Cgulp2ΔCgslx8Δ*, and *Cgulp2ΔCguls1Δ* double mutants.

From the TMT-labeled mass spectrometry, we have acquired 3281 proteins in total. 2194 proteins were retained for analysis after Principal Component Analysis (PCA) and eliminating proteins that displayed differences between biological replicates. Based on the abundance value of the proteins, we have calculated the p-value for each mutant using MeV_4.9.0 software and took a threshold of 0.05 for significant p-value. We identified 107, 87, 148, 278, and 300 proteins that were significantly upregulated and 90, 130, 241, 184, and 134 proteins were sig-nificantly downregulated in *Cgulp2Δ*, *Cgslx8Δ*, *Cguls1Δ* single mutants, *Cgulp2ΔCgslx8Δ*, and *Cgulp2ΔCguls1Δ* double mutants respectively (S2–S6 Tables). Next, we used the DAVID data-base to perform Gene Ontology (GO) to identify the pathways of the upregulated and downre-gulated proteins for each mutant in comparison with WT (S7–S11 Tables). The pathways are represented as horizontal bar graphs using SR plot software (Fig 6).

We have seen multiple pathways that were differentially affected in these mutants. These include proteins that are involved in metabolic processes, biosynthetic processes, membrane transport, ribosomal biogenesis, translation, proteasomal protein mediated degradation and others. Details of the protein components of pathways identified for each mutant is in S7–11 Tables. Proteomics data analysis revealed that the proteins involved in transport, particularly mitochondrial electron transport, transmembrane transport, and other ion transports were reduced in *Cgulp2Δ*, *Cgslx8Δ*, *Cguls1Δ* and *Cgulp2ΔCgslx8Δ*, but were higher in *Cgulp2ΔC-guls1Δ* double mutant (Figs 6 and S6).

We also observed proteins involved in ribosomal biogenesis and translation were reduced in the *Cgulp2Δ* and *Cgslx8Δ* single mutant. Similarly, a large group of mitochondrial and cyto-plasmic ribosomal proteins comprising both large and small subunits were reduced in *Cgulp2ΔCgslx8Δ* double mutant (Figs 6 and S7). Thus, the *Cgulp2ΔCgslx8Δ* double mutant is likely to be compromised in translation of both cytosolic and mitochondrial proteins leading to growth defects even at conducive temperatures in rich medium. Although the *Cgulp2ΔCguls1Δ* double mutant had increased levels of mitochondrial ribosomal proteins of both large and small subunits, the majority of cytoplasmic ribosomal proteins were reduced (Figs 6 and S7).

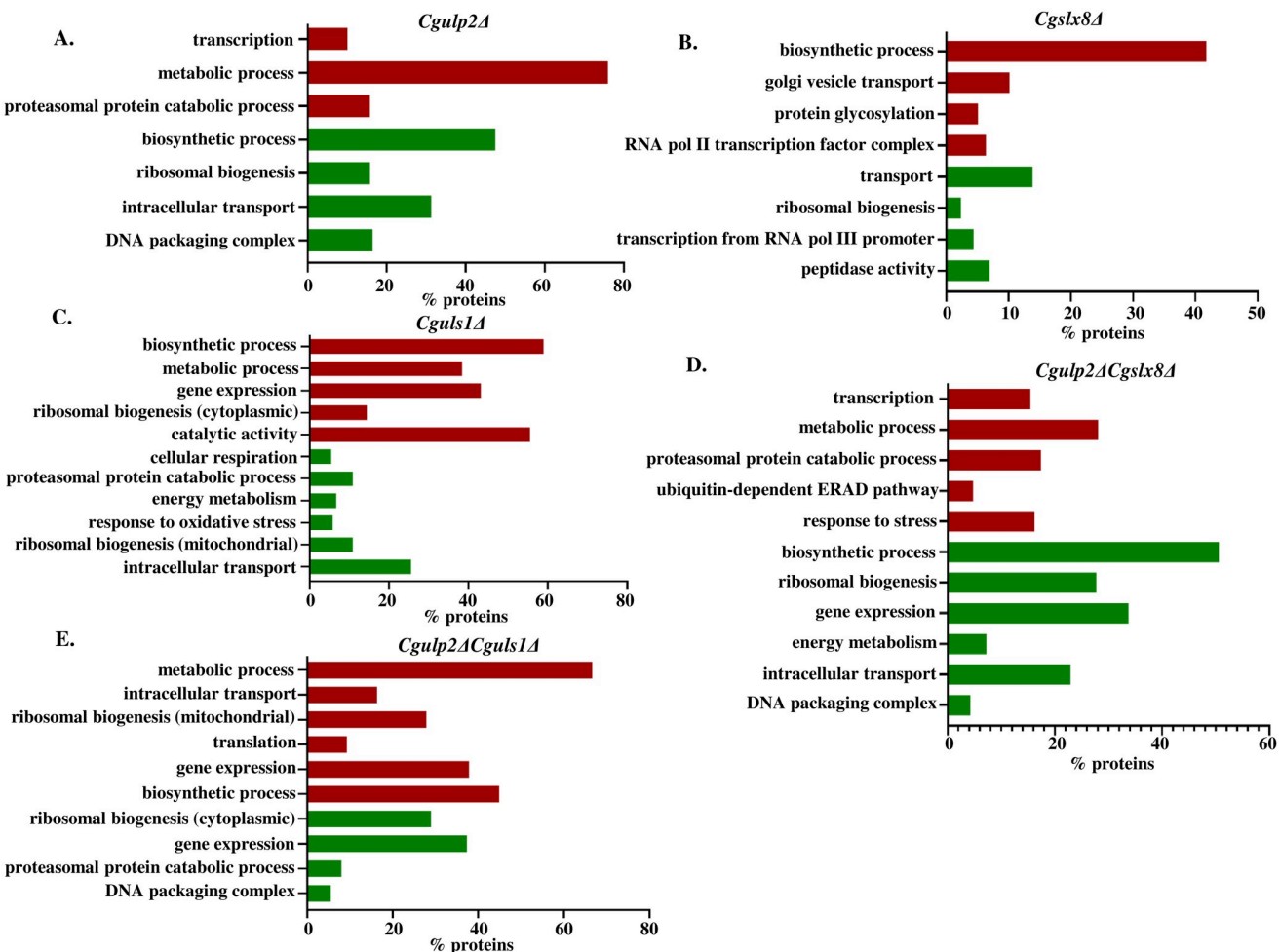

**Fig 6. Identification of differentially affected pathways for each mutant in comparison to WT.** The horizontal bar graph shows the pathways in **A)** *Cgulp2Δ*, **B)** *Cgslx8Δ*, **C)** *Cguls1Δ*, **D)** *Cgulp2ΔCgslx8Δ*, and **E)** *Cgulp2ΔCguls1Δ* mutant in comparison to WT. The red bar represents fraction of proteins in the indicated pathways with elevated protein levels and the green bar represents fraction of proteins in the indicated pathways with reduced protein levels.

This indicates members of the ribosomal biogenesis and translation pathway are more likely to be impacted by SUMO-mediated protein homeostasis.

Another interesting observation was that in *Cgulp2Δ* single mutant, there was elevated levels of a small number of proteins that belongs to the proteasomal degradation pathway. The majority of the proteins in the 26S proteasomal complex and those involved in endoplasmic reticulum-associated protein degradation (ERAD) which include CgKar2, CgCdc48, CgHlj1, CgHrd3, CgJem1, CgUbr1 and CgUfd1 were also highly upregulated in the *Cgulp2ΔCgslx8Δ* double mutant (Figs 6 and S8), suggesting that this enhanced levels of the degradative pathways could be responsible for the elevated levels of protein degradation occurring in this double mutant. In contrast, the *Cguls1Δ* and *Cgulp2ΔCguls1Δ* mutants showed downregulation of the ubiquitin-mediated proteasomal degradation pathway, possibly resulting in reduced protein degradation (Figs 6 and S8). This differential regulation of proteasomal pathway suggests that the balance of protein turnover was strongly influenced by the specific genetic context, particularly when combining deletion of CgUlp2 with either CgSlx8 or CgUls1.

These findings correlate with the SUMOylation profiles observed in the western blots (Fig 5) with *Cgulp2ΔCguls1Δ* strains with lowered proteasomal components showing higher levels of SUMO conjugates. Conversely, the *Cgulp2ΔCgslx8Δ* double mutants exhibit fewer SUMOylated proteins, consistent with increased proteasomal activity leading to elevated protein degradation. Moreover, MG132 treatment of the *Cgulp2ΔCgslx8Δ* mutants inhibits proteasomal degradation, leading to the accumulation of higher molecular weight SUMOylated proteins. Together, these observations highlight the link between proteasomal activity and SUMOylated protein regulation across these mutants.

As these deletion strains displayed altered growth and ability to combat stress, we examined the biosynthetic processes that are critical for growth. We found that several biosynthetic pathways were also differentially impacted in *Cgulp2Δ*, *Cgslx8Δ*, *Cguls1Δ* single mutants, *Cgulp2ΔCgslx8Δ*, and *Cgulp2ΔCguls1Δ* double mutants (Figs 6 and S9). Moreover, *Cgulp2ΔCgslx8Δ* also had lower levels of proteins involved in aminoacid, lipid, and ATP biosynthesis (S9 Fig, first column). Among the numerous biosynthetic processes, proteins involved in purine biosynthesis were lower in the *Cgulp2Δ* and *Cgulp2ΔCgslx8Δ* double mutant and higher in *Cgulp2ΔCguls1Δ* (Fig 7A). Nucleotide biosynthesis pathways fall into two categories: de novo pathways and salvage pathways (35,36). Proteins of de novo purine biosynthesis pathways were reduced in the *Cgulp2ΔCgslx8Δ* double mutant (Fig 7A). Purine nucleotide biosynthesis, as shown in Fig 7B, begins with 5-phosphoribosyl-1-pyrophosphate (PRPP), which is then converted through a sequence of reactions into inosine 5'-monophosphate (IMP). After that, IMP may go through various processes that produce either GMP or AMP, which are subsequently transformed into ADP and GDP, respectively [35,36]. Based on the mass spectrometry data, the *Cgulp2ΔCgslx8Δ* double mutant had downregulated levels of several proteins involved in this pathway. Particularly, CgAde5,7 was reduced in the *Cgulp2Δ* and *Cgulp2ΔCgslx8Δ* double mutant but was increased in the *Cgulp2ΔCguls1Δ* double mutant (Fig 7A).

We then investigated whether *CgADE5,7* overexpression could restore growth of the *Cgulp2ΔCgslx8Δ* double mutant at 30°C. In order to accomplish this, we created a construct with the *CgADE5,7* gene in the pGRB2.2 *C. glabrata* expression vector (carrying 3xFLAG at the N-terminus), and we used an α-flag antibody to confirm the expression of CgAde5,7 protein in the WT, *Cgulp2Δ*, *Cgslx8Δ*, *Cguls1Δ*, *Cgulp2ΔCgslx8Δ*, and *Cgulp2ΔCguls1Δ* mutants through western blot. All these mutants showed similar levels of expression of the CgAde5,7 protein (S11A Fig). In comparison to controls, the overexpression of the *CgADE5,7* gene resulted in some degree of growth restoration in the cells of the WT and mutants at 30°C and 42°C (Fig 7C). Significantly, we found that overexpressing the *CgADE5,7* gene allowed the *Cgulp2ΔCgslx8Δ* double mutant to partially recover its growth at 30°C and 42°C (Fig 7C–7E). Interestingly, the temperature sensitivity of the *Cgulp2ΔCguls1Δ* mutant at 42°C was also partly rescued by overexpressing CgAde5,7 indicating that while the CgAde5,7 levels were elevated at 30°C, it maybe compromised at 42°C in *Cgulp2ΔCguls1Δ*.

If the *Cgulp2ΔCgslx8Δ* and the *Cgulp2* were compromised for purine biosynthesis, they are likely to be more sensitive to purine biosynthesis inhibitors. We therefore, tested the sensitivity of the STUbL mutants to purine biosynthesis inhibitors, namely, azaserine, pemetrexed, and isatin. We examined the cell growth in WT and mutants. The initial, rate-limiting step involves the conversion of PPRP to PRA, which is catalyzed by PRPP amidotransferase. Azaserine, an antifungal agent, inhibits this step [35,36]. The anticancer drug, pemetrexed, prevents GAR transformylase from converting GAR to FGAR in the third step. The process by which the enzyme AIR carboxylase converts AIR to CAIR is inhibited by isatin, an antifungal agent.

In this study, we observed that the survival of *C. glabrata* WT cells decreased to approximately 50–60% when treated with azaserine, pemetrexed, and isatin, compared to the vehicle

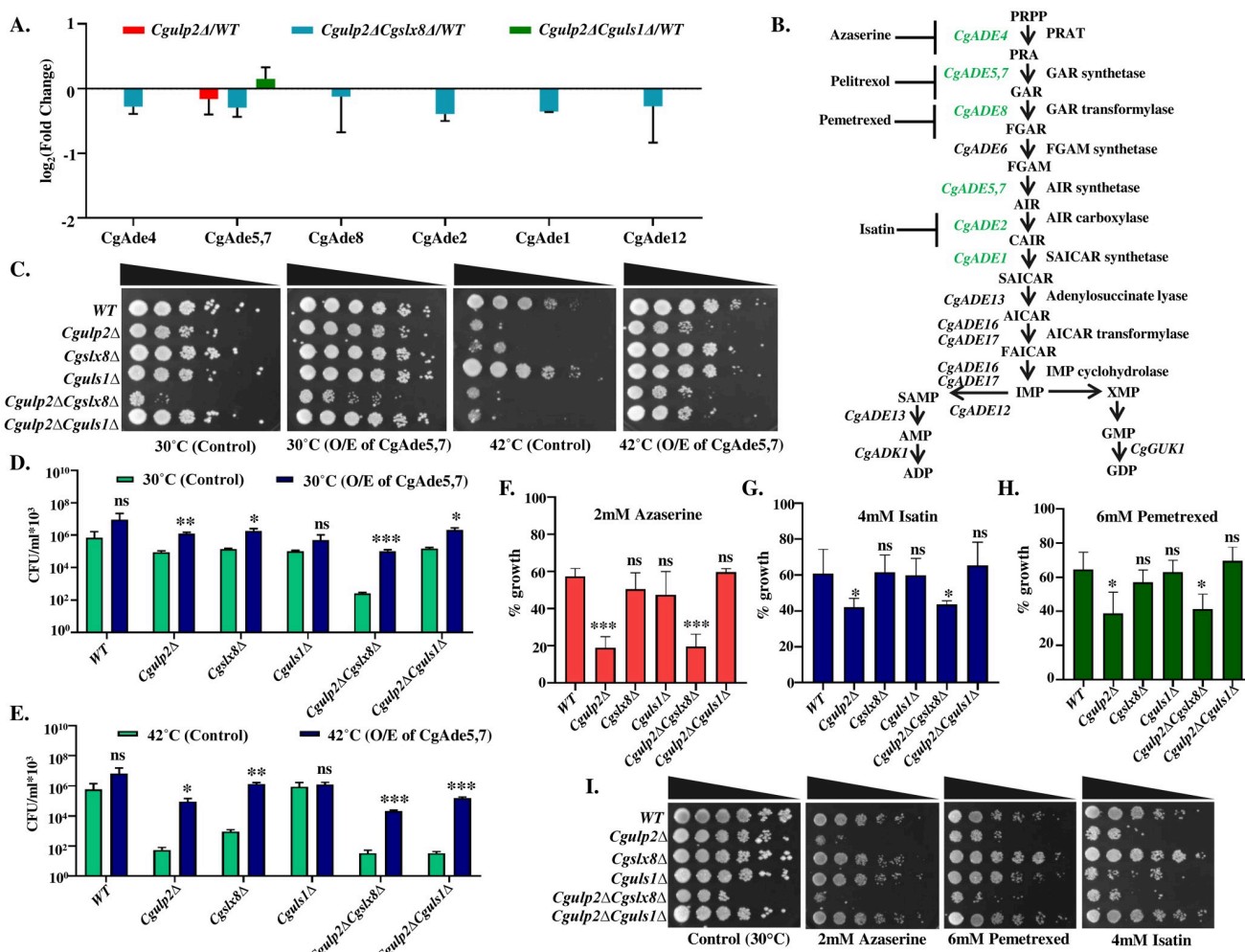

**Fig 7. Reduced growth of *Cgulp2ΔCgslx8Δ* double mutant is partly due to the altered levels of protein involved in the purine biosynthetic pathway. A.** Bar graph shows the levels of the proteins (in comparison to wild type) involved in this pathway for *Cgulp2Δ* (red bar), *Cgulp2ΔCgslx8Δ* (blue bars), and *Cgulp2ΔCguls1Δ* (green bar). The error bar represents the $\log_2$ standard deviation. **B.** de novo pathway of purine nucleotide biosynthesis. Proteins labeled in green color were obtained in the mass spectrometry analysis. The site of action of azaserine, pemetrexed, and isatin in the purine biosynthesis pathway are indicated. Abbreviation: PRPP- 5-phosphoribosyl-1- pyrophosphate, PRAT- phosphoribosylamidotransferase, PRA- phosphoribosylamine, GAR- glycineamideribonucleotide, FGAR- N-formylglycinamide ribonucleotide, FGAM- N-formylglycinamidine ribonucleotide, AIR- 5-aminoimidazole ribonucleotide, CAIR- AIR carboxamide, SAICAR- N-succino-5-aminoimidazole-4-carboxamide ribonucleotide, AICAR- aminoimidazole-4-carboxamide ribonucleotide, FAICAR- 5-formyl-AICAR, IMP, SAMP- adenylosuccinate, AMP- adenosine monophosphate, ADP- adenosine diphosphate, XMP- xanthosine monophosphate, GMP- guanosine monophosphate, GDP- guanosine diphosphate, ADK1-adenylate kinase 1, GUK1- guanylate kinase 1. **C.** Overexpression of *CgADE5,7* in STUbL mutants. pGRB2.2 plasmid encoding *CgADE5,7* was transformed in both the wild type (WT) and mutant strains of *C. glabrata* and the empty vector was used as control. To assess the impact of *CgADE5,7* overexpression on growth, 3 µl of 10-fold serial dilutions of exponential phase (1.0 A$_{600}$) grown cultures of *C. glabrata* strains were spotted in medium lacking uracil. Plate images were taken after 1–3 days of incubation at 30˚C and 42˚C, respectively. **D** and **E.** Bar graphs show the quantification of cell growth for each mutant upon overexpressing the *CgADE5,7* gene (blue bars), compared to the control strains (light green bars), at both 30˚C and 42˚C. Statistical significance was assessed using a two-tailed student's t-test (*- p ≤ 0.01; **- p ≤ 0.005; ***-p ≤ 0.0005; ns-non-significant). **F-H.** Growth assay in STUbL mutants. Overnight grown cultures of indicated *C. glabrata* strains were diluted in fresh YPD broth to an initial A$_{600}$ of 0.1. 100µl of diluted *C. glabrata* cell suspension were added to each well of a 96-well plate. Azaserine, pemetrexed, and isatin inhibitors were added in the concentration of 2mM, 6mM, and 4mM respectively to the cell suspension, and the final volume of 200µl for each well was maintained by adding YPD. YPD broth and cell suspension (100µl) without inhibitors were used as a control. Inoculum was incubated at 30˚C for 48 hr with 300rpm agitation. Absorbance at 600nm was measured at 48 hours to assess the survival rates, calculated relative to untreated control samples. The data are expressed as a percentage, with the no-drug well (vehicle control) of each strain set as 100%. Statistical significance was assessed using a two-tailed student's t-test (*- p ≤ 0.01; ***- p ≤ 0.0005; ns-non-significant). Data represent the mean of biological triplicates and the error bar represents the standard deviation. **I.** Growth of WT and mutants at 30˚C when treated with 2mM azaserine, 6mM pemetrexed, and 4mM isatin is shown. 2 µl of 5-fold serial dilutions of logarithmic phase grown culture of WT and mutants were spotted on YPD plates. Plate images were captured after 48 hours of incubation at 30˚C.

control. This reduction corresponded to a 40–50% growth inhibition, highlighting the significant impact of these purine biosynthesis inhibitors on WT cell growth (Fig 7F–7I). Specifically, survival for the *Cgulp2Δ* and *Cgulp2ΔCgslx8Δ* mutants dropped to 19% following treatment with 2 mM azaserine, indicating an 81% inhibition in these strains. Furthermore, survival was reduced to 40% with 6 mM pemetrexed and 42% with 4 mM isatin. In contrast to the WT, these mutants exhibited notably slower growth, especially under 2 mM azaserine treatment (S11B Fig). The *Cgslx8Δ*, *Cguls1Δ*, and *Cgulp2ΔCguls1Δ* mutants displayed growth and survival patterns similar to those of the WT (Fig 7F–7I). Altogether, these data suggest that compromised purine nucleotide biosynthesis was one of the key reasons for the slow growth of the *Cgulp2ΔCgslx8Δ* mutant at 30˚C and the restoration of cell growth in *Cgulp2ΔCguls1Δ* double mutant. This suggests that particular components and pathways are more sensitive to the SUMO-mediated protein homeostasis.

## Mitochondria function is compromised in *Cgulp2Δ* and *Cgulp2ΔCgslx8Δ*

Apart from the biosynthetic processes, proteins involved in mitochondrial protein synthesis and members of the electron transport chain were affected in the *Cgulp2ΔCgslx8Δ* double mutant. Complexes II, III, IV, and V were highly reduced in *Cgulp2ΔCgslx8Δ*, suggesting that mitochondrial metabolism was severely affected. In the case of *Cgulp2Δ*, there was no clear down regulation, with few of components being present in higher or lower levels than wild type. In addition, proteins associated with these complexes were either upregulated or up to the level of WT in the *Cgulp2ΔCguls1Δ* double mutant (Fig 8A). Dysregulation of mitochondria has several consequences such as ATP depletion, ROS generation, and oxidative stress [37,38]. Also, *Cgulp2Δ* and *Cgulp2ΔCgslx8Δ* have reduced growth in non-fermentable carbon sources like 2% ethanol, 3% glycerol, and 2% lactate, and *Cgulp2ΔCguls1Δ* appeared to suppress the defects seen in the single mutant of *Cgulp2Δ* (Fig 1D).

We initially examined the mitochondrial morphology of the wild type and mutants of the *C. glabrata*. The wild type cells exhibited a tubular mitochondrial structure, while *Cgulp2ΔCgslx8Δ* displayed 90% fragmented mitochondria, followed by the *Cgulp2Δ* mutant which had 60% fragmentation. However, *Cgulp2ΔCguls1Δ* mutant had a mitochondrial morphology similar to that of the wild type (Fig 8B). To assess if the mitochondrial morphological changes in the mutants influenced reactive oxygen species (ROS) levels, we stained WT and *C. glabrata* mutants with MitoSOX, a mitochondria superoxide indicator. *Cgulp2ΔCgslx8Δ* cells showed elevated ROS levels, with a stronger fluorescent signal followed by *Cgulp2Δ* mutant. *Cgulp2ΔCguls1Δ* and WT had no or very little amount of ROS (Fig 8B). Further, to validate that the increased signal in *Cgulp2Δ* and *Cgulp2ΔCgslx8Δ* cells was due to enhanced mtROS production, we treated these cells with N-acetylcysteine (NAC), an antioxidant known to reduce ROS generation. NAC treatment significantly decreased fluorescent signals in both *Cgulp2Δ* and *Cgulp2ΔCgslx8Δ* cells compared to untreated cells (Fig 8C). These data together suggest a compromised mitochondrial function in *Cgulp2ΔCgslx8Δ* and *Cgulp2Δ*. To further confirm this data, we also grew cells in the presence of NAC and found that growth of the *Cgulp2Δ* single mutant improved on NAC medium, *Cgslx8Δ* single mutant displayed growth similar to WT under both conditions, and the *Cgulp2ΔCgslx8Δ* double mutant showed significantly improved growth with NAC relative to growth without NAC (Fig 8D–8E). These findings were further supported by the doubling time analysis, which showed a consistent trend with the growth curves, indicating a reduced doubling time for *Cgulp2Δ* and *Cgulp2ΔCgslx8Δ* in the presence of NAC (Fig 8F–8G). After NAC treatment, 29% of *Cgulp2ΔCgslx8Δ* cells displayed a restored tubular mitochondrial structure, and the *Cgulp2Δ* mutant showed minor improvement, with 65% of cells exhibiting tubular morphology (Fig 8C). These data establish

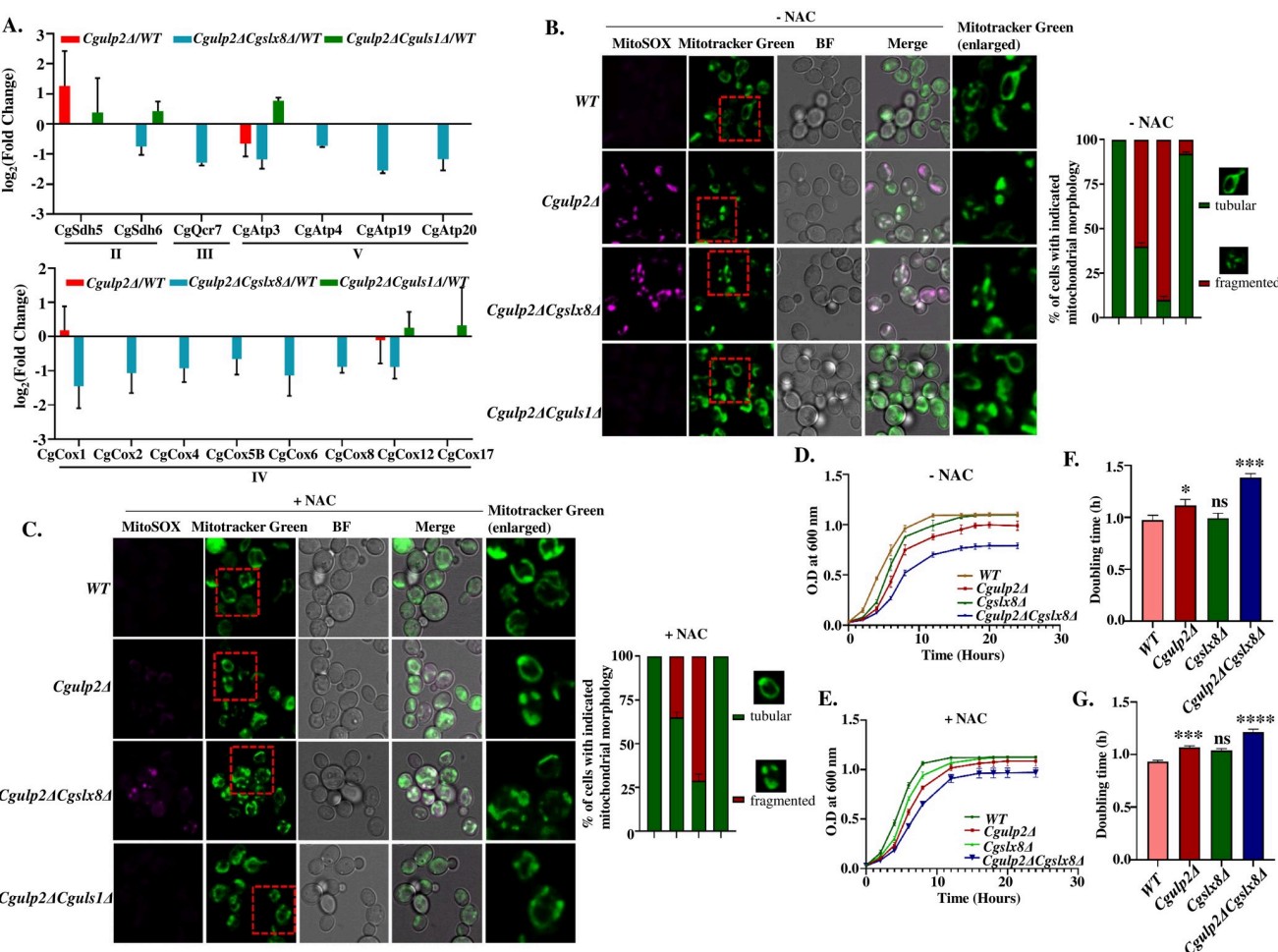

**Fig 8. Mitochondrial functions are affected in *Cgulp2Δ* and *Cgulp2ΔCgslx8Δ*. A.** Bar graph shows the differential levels of the proteins involved in the mitochondrial complexes of the ETC pathway in *Cgulp2Δ*, *Cgulp2ΔCgslx8Δ*, and *Cgulp2ΔCguls1Δ*. The error bar represents the standard deviation. **B.** Cells were grown in YPD medium without 10mM NAC (- NAC) at 30°C and stained with MitoTracker Green and MitoSOX (magenta). Confocal microscopic images (left panel) of WT, *Cgulp2Δ*, *Cgulp2ΔCgslx8Δ*, and *Cgulp2ΔCguls1Δ* to check mitochondrial morphology and ROS levels. The scale bar represents 5 μm. The right panel displays the quantification of mitochondrial morphology in WT and mutants in absence of NAC. The red dotted box indicates the region enlarged for better visualization. **C.** Cells were grown in YPD medium with 10mM NAC (+ NAC) at 30°C for 12 hours and then stained with MitoTracker Green and MitoSOX. Images were captured in a confocal microscope (left panel). Examples of tubular and fragmented mitochondria selected for quantification are shown. The scale bar is 5 μm. The right panel shows the quantification of mitochondrial morphology in WT and mutants following NAC treatment. For each experiment, 100 cells were counted, and with three independent replicates, a total of 300 cells were analyzed. **D-E.** Overnight grown cultures of indicated *C. glabrata* strains were inoculated in YPD medium without NAC (- NAC) or with 10mM NAC (+ NAC) at an initial $A_{600}$ of 0.1. Absorbance at 600nm was measured at regular intervals. Data represent the mean of biological triplicates. **F-G.** Doubling time was calculated for the *C. glabrata* strains with and without NAC and plotted. Statistical significance was assessed using a two-tailed student's t-test (*- p ≤ 0.01; ***- p ≤ 0.0005; ****- p ≤ 0.0001; ns-non-significant).

that increased ROS levels could be one of the factors that leads to compromised growth of the *Cgulp2ΔCgslx8Δ* double mutant.

## Discussion

In this work, we show that in *Candida glabrata*, protein homeostasis is regulated by protein deSUMOylase CgUlp2 and the STUbLs. A balance of the removal of polySUMOylation by CgUlp2 and degradation of polySUMOylated proteins by STUbLs is critical for growth,

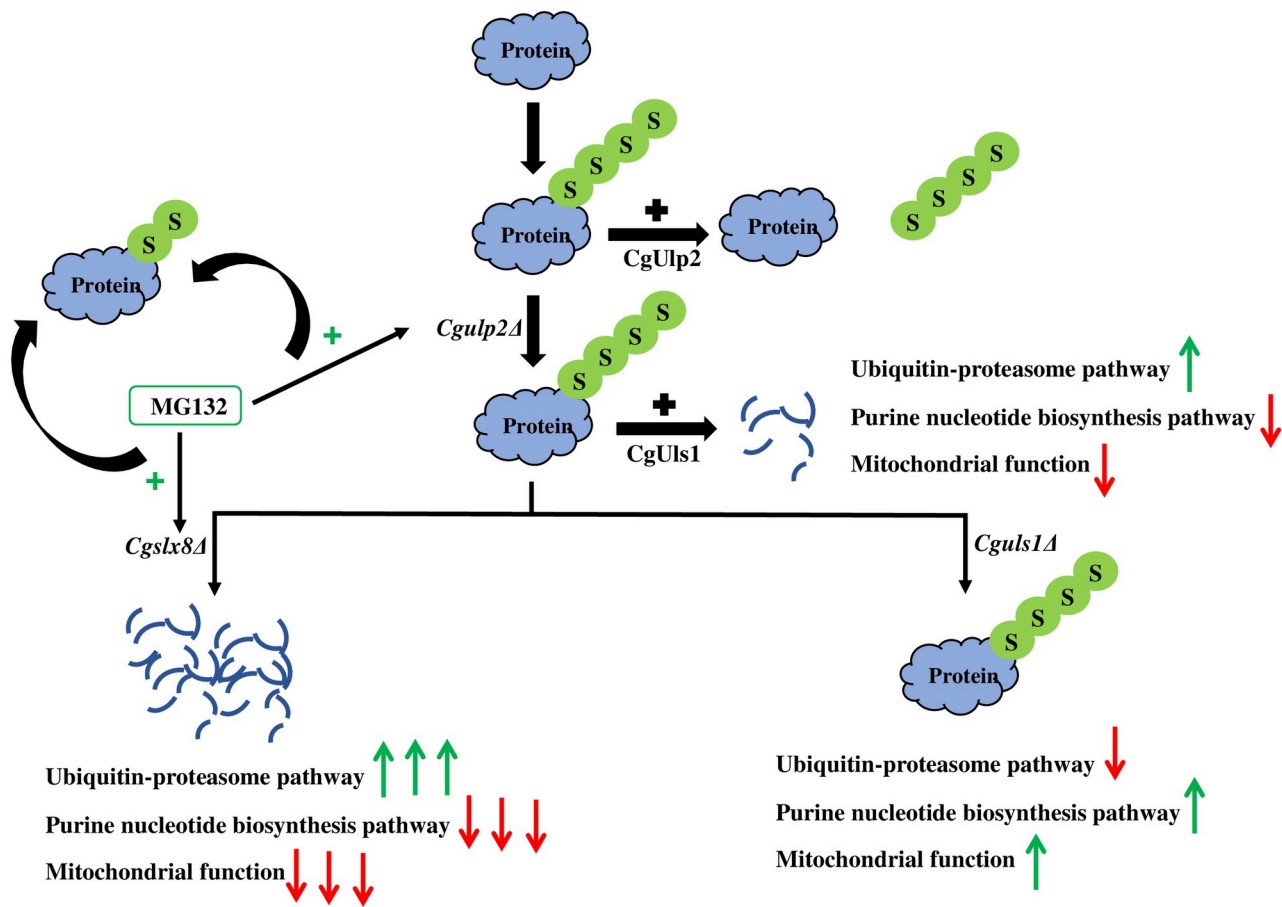

**Fig 9. Molecular basis for phenotype observed in *Cgulp2Δ* and STUbLs in *C. glabrata*.** In the presence of CgUlp2, SUMO chains are removed from the target protein whereas, in its absence, polySUMOylated proteins are accumulated and targeted for protein degradation via STUbL, primarily CgUls1. This is a key contributor to the homeostasis of the target proteins. As more polySUMOylated protein accumulates in the absence of CgUlp2, in *CgulsΔ*, it is not degraded. The double mutant *Cgulp2ΔCgslx8Δ* exhibits an increased level of protein degradation possibly because of upregulated activity of CgUls1. Following MG132 treatment, degradation of proteins was prevented and SUMOylated proteins appeared in *Cgulp2Δ* and *Cgulp2ΔCgslx8Δ* mutants and so did the improved growth. Based on the experimental validation of quantitative mass spectrometry data, ubiquitin-proteosome pathway, purine nucleotide biosynthesis pathway, and mitochondrial function were found to be significantly impacted in *Cgulp2ΔCgslx8Δ* double mutant. This was followed by the *Cgulp2Δ* single mutant. We propose that these are the primary causes of the increased protein loss and slow growth of *Cgulp2ΔCgslx8Δ* cells at 30°C. The ubiquitin-proteosome pathway components were decreased whereas purine nucleotide biosynthesis pathway and mitochondrial proteins had increased in *Cgulp2ΔCguls1Δ* due to which restoration of growth and accumulation of polySUMOylated proteins was seen in this mutant compared to the *Cgulp2Δ* mutant.

combating stress and for *C. glabrata* to survive in host cells (Fig 9). We had earlier identified the potential STUbLs in *C. glabrata* based on sequence similarity with STUbLs of *S. cerevisiae*. While the full length deletions of individual STUbLs renders the organism susceptible to several stress causing agents and inability to proliferate in macrophages, it is the combination of STUbL deletions with the deSUMOylase CgUlp2 that has profound phenotypes. The loss of *Cguls1* along with the *Cgulp2* suppresses some of the stress sensitivity phenotypes of single mutant *Cgulp2*. Conversely, the loss of *Cgslx8* along with *Cgulp2* exacerbates the growth defects of *Cgulp2* even at conducive temperatures. This intriguing phenotype prompted us to investigate the connections between STUbLs and CgUlp2.

When we examined the overall SUMO proteome of *Cgulp2Δ*, Cg*slx8Δ*, *Cguls1Δ* and the double mutants *Cgulp2ΔCgslx8Δ* and *Cgulp2ΔCguls1Δ*, there are very stark differences. Of

particular importance is that the *Cgulp2Δ* cells have fewer SUMOylated proteins, especially the higher molecular weight proteins, which could be the polySUMOylated proteins. While this appears counter intuitive at first, as we conjectured earlier and demonstrate here, this is because the polySUMOylated proteins are targeted for degradation, leading to increased protein degradation in the absence of CgUlp2. When STUbLs were deleted, we expected to see restoration/accumulation of these polySUMOylated proteins. Interestingly, we find that this is partially restored in the *Cgulp2ΔCguls1Δ* but there is increased protein degradation in *Cgulp2ΔCgslx8Δ* double mutants (Fig 9). This suggests that CgUls1 could be a major STUbL in *C. glabrata* for the CgUlp2 targets. The observation that *Cgulp2ΔCgslx8Δ* is more susceptible to stress than the *Cgulp2Δ* or *Cgslx8Δ* single mutant suggested increased protein degradation in this mutant. We confirmed this by SUMO western blots, MG132 treatment and also uncovered that CgUls1 was upregulated in the absence of Cg*slx8*. However, this is not a global protein degradation as levels of several proteins in *Cgulp2ΔCgslx8Δ* are similar to wild type and some proteins, especially those involved in the metabolic processes, are even present at a higher level than wild type (S10 Fig). This indicates that targeting the SUMO-mediated protein homeostasis pathway will selectively incapcitate a few critical pathways of the pathogen.

The contrasting phenotypes of the two double mutants provided a handle to investigate the downstream pathways that are likely to be targets of these STUbLs. Our quantitative whole proteome analysis showed that several pathways were affected (Fig 6). Of note, we found that the components of protein degradation pathways including 23S proteasome and ERAD pathway were significantly higher in *Cgulp2ΔCgslx8Δ* and some of these were high in the *Cgulp2Δ* single mutant also. In contrast, these components were present at lower than wild type levels in the *Cgulp2ΔCguls1Δ* double mutant (S8 Fig). This could be the reason for the increased protein degradation observed in *Cgulp2ΔCgslx8Δ* and *Cgulp2Δ*, and also supports our observations with MG132 restoring growth to these mutants. Components of the cytosolic ribosome biogenesis are reduced in almost all mutants except for a few components that are higher in *Cguls1Δ* (Figs 6 and S7). This suggests that while there may be global reduction in translation when the SUMO pathway is perturbed, it is the increased levels of the protein degradation pathways in the *Cgulp2ΔCgslx8Δ* mutant that leads to reduced levels of proteins that were polySUMOylated.

We have also dissected the potential downstream pathways that are affected by altered proteostasis and establish that both purine biosynthesis and mitochondrial function are affected in the absence of CgSlx8. This correlates well with the phenotypes observed in the mutants. Metabolic processes especially nucleotide synthesis is critical for combating stresses like DNA damage. The *Cgulp2ΔCgslx8Δ* and *Cgulp2Δ* mutants are unable to effectively use non-fermentable carbon sources like lactate and glycerol. There is also increased ROS in these mutants even under unperturbed conditions (Fig 8B). This indicates poor mitochondrial health in these mutants. Mitochondria are important for fungal pathogenesis as the organism adapts to establish infection in mammalian cells [39,40]. The compromised mitochondrial function could be one of the factors for reduced capacity of these mutants to survive and multiply in macrophages.

Our studies confirm that SUMO-mediated ubiquitin ligases are crucial for protein homeostasis and even more so during infection. A significantly reduced survival in the macrophages by all the STUbL mutants, compared to *Cgulp2* at 2 hours post infection is a clear indication of their inability to survive in the phagosomes. Probably, a reduced mitochondrial function, low pH inside the phagosome and compromised nucleotide synthesis together contribute to the inability to grow and survive by these mutants in the macrophages. Interestingly, mutants *Cgulp2Δ* and *Cgulp2ΔCgslx8Δ* grow well in YPD but not in RPMI media, which is closer to the mammalian cell milieu than YPD (S4 Fig). This shows that compromising the STUbL pathway

may be more detrimental than estimated and testing YPD is not reflective of the requirements of the pathogen to establish a successful infection. Our macrophage infection studies clearly show that the STUbL mutants cannot survive and proliferate in macrophages. *C. glabrata* is one of the few pathogens that not only survives but also proliferates in macrophages; therefore, the inability of STUbL mutants to replicate in macrophages is a good indicator of the importance of STUbLs, and by extrapolation, SUMO mediated protein homeostasis, in establishing successful infection by *Candida glabrata* [41].

While several functions of the deSUMOylase and STUbLs are similar in *C. glabrata* compared to other model organisms studied, there are some key differences, even between *C. glabrata* and *S. cerevisiae* which are closely related. Ulp2 plays a critical role in regulating SUMO dynamics, genome stability, and protein homeostasis in many organisms. In *C. glabrata*, deletion of *CgULP2* results in sensitivity to oxidative stress and DNA damage, as well as defects in virulence, adherence, and biofilm formation [25]. This mirrors the phenotype observed in *S. cerevisiae*, where *ulp2Δ* mutants display increased sensitivity to genotoxic agents such as methyl methanesulfonate (MMS) and hydroxyurea (HU) due to impaired DNA repair and protein homeostasis, and chromosomal instability [17,42,43]. Similarly, in other organisms including *S. pombe*, *C. albicans* and humans (SENP6 and SENP7), loss of Ulp2 leads to genome instability, reduced DNA repair and aneuploidy [44–47]. However, unlike *S. cerevisiae*, *Cgulp2Δ* doesnot accumulate higher molecular weight SUMO conjugates.

Previously, we identified SUMO-targeted ubiquitin ligases (STUbLs), including Slx5, Slx8, and Uls1, across various fungi, with varying degrees of conservation. Slx5 was found to be the least conserved, present only in the Saccharomycetes class of the Ascomycota phylum, whereas Slx8 and Uls1 showed broader conservation across fungi [30]. In *S. cerevisiae*, Slx5 and Slx8 form a heterodimeric complex that is critical for maintaining genome stability and deletion of *SLX5* or *SLX8* results in sensitivity to agents like MMS and HU [44,48–50]. However, in *C. glabrata*, deletion of these genes individually and in combination does not result in sensitivity to DNA damage-inducing agent and the individual deletions display distinct phenotypes. *Cgslx5Δ* are not temperature sensitive, and surprisingly, suppress the temperature sensitivity of *Cgslx8Δ*. In addition, the *Cgslx5ΔCgslx8Δ* strains are unable to grow on ethanol, and intriguingly, loss of *Cgslx5* individually or in combination with *Cgslx8Δ* restores some of the higher molecular weight SUMO conjugates. Slx5 is not present in all organisms that contain Slx8, suggesting that Slx8 can independently carry out the STUbL activity [30]. Thus it appears that there is functional separation of Slx5 in *C. glabrata* (and perhaps in other fungi as well) and loss of Slx5 in other organisms. Uls1 plays a key role in regulating replication stress and maintaining rDNA stability in *S. cerevisiae*. Although *uls1* mutant cells do not exhibit sensitivity to DNA damaging agents, they accumulate high molecular-weight SUMO conjugates [51–53]. In contrast, deletion of *ULS1* in *C. glabrata* does not lead to any observable phenotypic response under stress conditions, nor does this strain accumulate high molecular-weight SUMO conjugates, indicating potential functional differences between the species.

A key difference between *C. glabrata* and *S. cerevisiae* that has come out of this study is the SUMO/STUbL/proteasome axis. In *S. cerevisiae*, *Δulp2*, *Δslx5/slx8* and *Δuls1* all accumulate polySUMO proteins [16,53]. While in many studies specific Ulp2 substrates have been identified that are ubiquitylated and degraded, in *Δulp2* overall SUMO proteome is higher, especially, polySUMO proteome [20]. This suggests that they are not targeted to the proteasome for degradation effectively. However, in *C. glabrata*, in all these mutants, we see reduced higher molecular weight SUMO conjugated proteins and increased degradation of proteins. This suggests that, in *C. glabrata*, the SUMO-STUbL-proteasome axis is more tightly coupled and plays a key role in overall protein homoeostasis. Our mass spectrometry and MG132 experiments corroborate with this conclusion.

The study presented here suggests SUMO-meditated protein homeostasis pathway could be a good target for antifungal development. While STUbLs have not been studied in pathogenic fungi, the SUMOylation pathway is conserved across fungi. SUMO is either essential for survival (*S. cerevisiae* and *C. glabrata)* or for stress response and/or virulence (all other fungi studied) [26–30,47,54–61]. Understanding the SUMO-dependent protein homeostasis pathway in pathogenic fungi, will widen our understanding of fungal pathogenesis in general and also uncover new targets for antifungal therapies. Targeting this process provides a possibility of identifying several downstream targets as it strikes multiple pathways simultaneously. Our data also provide an overview of all potential pathways that are affected by perturbing the SUMO-mediated protein homeostasis in *C. glabrata*, offering an additional approach for designing combinatorial therapies. For instance, some of the purine biosynthesis inhibitors are already used as antifungals in the clinic [62–64]. If this can be combined with the inhibitors of Ulp2, they could augment the effect of these drugs. Therefore, further understanding the pathways that are regulated by SUMO-mediated protein homeostasis in establishment of successful infection would enable the design of better antifungals.

## Methods

**Bacterial and yeast culture conditions**- The LB medium, which contains 1% tryptone, 0.5% yeast extract, and 1% NaCl, and the YPD medium, which contains 1% yeast extract, 2% peptone, and 2% dextrose, were used to routinely maintain the strains. *C. glabrata* cells were grown in either YPD or CAA medium (0.6% casamino acids, 0.67% yeast nitrogen base (YNB) without amino acids, and 2% dextrose) for uracil phototrophy. *C. glabrata* cells in the logarithmic phase were obtained by incubating overnight-grown cultures in fresh YPD/CAA media for roughly 4–6 hours at 30˚C. Strains, and plasmids used in this study are listed in Table 1.

**Gene disruption and cloning in *Candida glabrata***—In *C. glabrata*, ORFs were disrupted using homologous recombination with a nourseothricin resistance cassette. This cassette included the nourseothricin acetyltransferase (*NAT1*) gene, bordered by the 5'- and 3'-UTR regions of the target gene, each approximately 600 bp long, amplified from wild-type genomic DNA. The *NAT1* gene was amplified from a plasmid and fused with parts of the UTRs, creating PCR products with overlapping 300–350 bp regions. These fused PCR products were co-transformed into the wild type strain, followed by plating on YPD medium. After 12–16 hours of incubation at 30˚C to allow for homologous recombination, the cells were replica-plated onto YPD with 200 mg/ml nourseothricin and incubated for another 24 hours. Colonies that grew were screened via PCR to confirm gene disruption through homologous recombination [67].

The parental single-deletion strain was transformed with the *FLP1* flippase plasmid and selecting transformants based on uracil prototrophy in the CAA medium in order to generate double-deletion strains. The *NAT1* cassette was excised by the flippase enzyme at FRT (Flippase recognition target) sites flanking the *NAT1* gene [68]. The transformants that were sensitive to nourseothricin were isolated and *FLP1* flippase plasmid was removed after four to five passages in a nonselective YPD medium which was further verified in CAA plates. These cells were then transformed with a PCR-amplified linear segment from the genomic DNA of a single deletion strain and carried the sequence 5′-untranslated region (5′-UTR)-*NAT1*cassette-3′-untranslated region (3′-UTR). Colonies that were resistant to nourseothricin were isolated, and a PCR was used to check for gene disruption for double deletion strain [69].

To create complemented strains, the genes for *CgSLX5* (1.5 kb), *CgSLX8* (633bps), and *CgULS1* (4.2 kb) were amplified from wild-type genomic DNA using the high-fidelity Vent polymerase. These genes were then inserted into the pGRB2.2 (CKM 364) plasmid at the

**Table 1. List of strains and plasmids used in this study.**

| Yeast strain | Genotype | Reference |
|---|---|---|
| YRK19 | *ura3Δ::Tn903 G418^R (WT Ura- strain)* | [25,65] |
| YRK990 | *YRK 19 except Cgulp2Δ::nat1* | [25] |
| KRC6 | *Smt3::hyg/pCKM405* | [25] |
| KRC30 | *YRK 19 except Cgslx8Δ::nat1* | This study |
| KRC33 | *YRK 19 except Cguls1Δ::nat1* | This study |
| KRC36 | *YRK 19 except Cgulp2Δ::nat1/CKM 687* | This study |
| KRC42 | *YRK 19 except Cgulp2ΔCguls1Δ::nat1* | This study |
| KRC45 | *YRK 19 except Cgslx5Δ::nat1* | This study |
| KRC46 | *YRK 19 except Cgslx5ΔCguls1Δ::nat1* | This study |
| KRC47 | *YRK 19 except Cgslx5ΔCgslx8Δ::nat1* | This study |
| KRC50 | *YRK 19 except Cgslx5Δ::nat1/CKM 687* | This study |
| KRC53 | *YRK 19 except Cgulp2ΔCgslx8Δ::nat1* | This study |
| **Plasmid** | **Description** | **Reference** |
| CKM 364 | pGRB2.2, a CEN-ARS plasmid of *C. glabrata* carrying *S. cerevisiae URA3* as a selection marker. MCS sites are flanked by *S. cerevisiae* PGK1 promoter at one end and by 3′ UTR of *HIS3* at the other end. | [65] |
| CKM 382 | *CgULP2* (2.7 kb) inserted in BamHI-XhoI sites of CKM 364 | [25] |
| CKM 402 | pCN-PDC1, a high expression promoter with *nat1* gene | [25] |
| CKM 405 | 6XHIS3XFLAG tagged *CgSMT3* in CKM402 | [25] |
| CKM 468 | 6XHIS3XFLAG tagged *CgSMT3* in CKM364 | [25] |
| CKM 476 | 3XFLAG tagged *CgSMT3* in CKM 364 | [25] |
| CKM 686 | NAT cassette cloned in pCR2.1 plasmid | [66] |
| CKM 719 | *CgSLX5* (1.5 kb) inserted in BamHI-XhoI sites of CKM 364 | This study |
| CKM 720 | *CgSLX8* (633 bps) inserted in BamHI-XhoI sites of CKM 364 | This study |
| CKM 741 | *CgULS1* (4.2 kb) inserted in BamHI-XhoI sites of CKM 364 | This study |
| CKM 758 | 6XHIS3XFLAG tagged *CgULS1* under its native promoter in CKM 364 | This study |
| CKM 794 | 3XFLAG tagged *CgADE5,7*, replace CgSMT3 from CKM 476 and inserted *CgADE5,7* with BamHI/XhoI sites in frame with 3XFLAG in pGRB2.2 vector | This study |

BamHI-XhoI sites downstream of PGK1 promoter. Plasmids were sequenced for confirmation. For the complementation assay, the resulting recombinant plasmids were expressed in the appropriate deletion mutants.

**Plate growth assay**- *C. glabrata* wild type and deletion strains were grown in YPD/CAA medium at 30°C until reaching an O.D of 1.0 at $A_{600}$. Then 10-fold serial dilutions were made in 96 well plates. 2–3μl of diluted cell suspensions were spotted onto appropriate plates and incubated for 2–4 days at the required temperatures to evaluate cell growth [25].

**Antifungal susceptibility test**- The *C. glabrata* wild type (WT) strain and mutants (*Cgulp2Δ*, *Cgslx8Δ*, *Cguls1Δ*, *Cgulp2ΔCgslx8Δ*, and *Cgulp2ΔCguls1Δ*) were cultured in YPD broth at 30°C. Overnight cultures were diluted to an $OD_{600}$ of 0.1. Fluconazole, caspofungin, and amphotericin B were prepared at stock concentrations of 2 mg/ml, 5 μg/ml, and 250 μg/ml, respectively, using sterile distilled water. WT cells were treated with different concentrations of fluconazole (16, 32, and 64 μg/ml), caspofungin (75, 150, and 225 ng/ml), and amphotericin B (5, 10, and 20 μg/ml). Growth inhibition was assessed by measuring $OD_{600}$ after 48 hours of drug exposure. The concentration at which 50% growth inhibition occurred (IC50) was determined for each antifungal agent and identified as 32 μg/ml for fluconazole, 150 ng/ml for caspofungin, and 10 μg/ml for amphotericin B. The mutants were exposed to 32 μg/ml fluconazole, 150 ng/ml caspofungin, and 10 μg/ml amphotericin B for 48 hours. $OD_{600}$ was measured at

48 hours to assess the survival. All experiments were performed in triplicates. The survival was calculated relative to untreated control samples.

**Western blot**- Using trichloroacetic acid (TCA) precipitation, whole-cell extracts from the logarithmic phase (absorbance at $A_{600}$-1) *C. glabrata* cultures were prepared. Protein extraction was conducted using equal number of cells as determined by absorbance. Protein was separated on SDS-PAGE was followed by transfer onto PVDF membrane. After blocking for 1 hour with 5% skim milk powder, the membrane was incubated with α-flag primary antibody (Sigma, 1:10,000) for 14–16 hours. Following washes with TBST, blots were incubated with mouse HRP-conjugated secondary antibody (Abcam, 1:20,000) for 1 hour. Blots were developed using G-Biosciences chemiluminal development solutions after washes with TBST. Blots were further reprobed with α-tubulin antibody as the loading control.

**Preparation of THP-1 Macrophage monolayer**- Human monocyte, THP-1 cells were seeded in a 25mm culture flask at 37˚C, in a humidified incubator with 5% CO2, in RPMI-1640 culture medium, supplemented with 10% FBS, 2 mM glutamine, and antibiotic cocktail. Cells were allowed to grow until a monolayer was formed. Cell density was maintained at 2.5 x $10^5$ cells/ml by counting with a hemocytometer and 1μl of 160mM PMA stock (phorbol 12-myristate 13-acetate solution prepared in DMSO) was added to 10ml THP-1 cell suspension (final concentration 16nM) to stop the cells from further division and to convert them into macrophages. After 12 hours, the old medium containing PMA was removed and a pre-warmed fresh RPMI-1640 complete medium was added and allowed the cells to recover for another 12 hours at 37˚C. The appearance of macrophage cells were checked under an inverted microscope for their flattened, spindle-shaped, and adherent morphology.

**Infection of *C. glabrata* in THP-1 cells**- *C. glabrata* strains were grown in liquid YPD media at 30˚C with 180 rpm of agitation. A cell density of 2.5 x $10^6$ cells/ml was used to infect the cells, maintaining a multiplicity of infection of 1:10. *C. glabrata* cells were co-cultured with THP-1 macrophages for 2 hours, followed by gently washing several times (3–6 times) using 1 ml of sterile PBS to ensure the complete elimination of the non-phagocytosed yeast pathogen. Cells were then incubated for 8, 12, and 24 hours to examine the *C. glabrata* cell survival andproliferation.

To quantify internalized yeast, infected macrophages were washed and lysed using 1ml of sterile water for 2–3 minutes in the culture plate. Cells were scraped softly and cell lysate was collected. The cells were then serially diluted to 100 fold and 100μl of the suspension was plated on a YPD agar followed by incubation at 30˚C for 1–2 days. Colonies that appeared on YPD plates were counted manually, and CFU (colony forming units) was calculated.

**Biofilm formation**- Assay for biofilm was performed on flat-bottomed 24-well polystyrene plates. YPD-grown *C. glabrata* cells in the exponential phase were added to the plate and incubated at 37˚C with mild rotation (75 rpm) for 90 minutes. Wells were then washed twice using PBS, followed by the addition of 1ml of RPMI-1640 complete media to each well. The plate was then incubated at 37˚C with 75 rpm, for another 24 hours to promote biofilm formation. The next day, 500μl of used media was pipetted out and fresh 500μl RPMI containing 10% FBS was added. The plate was further incubated for 48 to 72 hours. During incubation, every day 500μl of fresh media was replaced to provide continuous nutrients for the growing biofilm. After 72 hours, the plate was washed with PBS to eliminate all the planktonic cells. Cells were then collected in PBS by scraping, pelleted down, and stored at -80˚C for RNA isolation.

**XTT assay**- XTT assay was used to quantify biofilm formation in wild type and various deletion strains of *C. glabrata*. After 72 hours of biofilm formation, the plate was washed thrice with PBS to remove unattached cells. 100μl of 1x XTT solution was added, and the plate was

incubated in the dark for one hour. Absorbance at 492nm was measured, RPMI-1640 complete medium serving as a negative control, and the absorbance of the *Candida*-grown wells was subtracted from this value [70,71].

## Quantitative Real-Time PCR

*C. glabrata* cells were grown overnight in YPD at 30˚C, it was sub-cultured till $OD_{600}1$. Cells were treated with DNA damaging agent (0.04% MMS), and hydroxyurea (100mM) and allowed to grow for additional 2 hours. For temperature sensitivity analysis, *C. glabrata* cells were shifted to 42˚C and incubated for 2 hours. The cultures were then pelleted down and stored at -80˚C for RNA extraction. For RNA extraction, the Ribo Pure kit from Invitrogen was used, followed by cDNA conversion using the SuperScript III First-strand Synthesis System for RT-PCR (Invitrogen) with 1 µg RNA as a template. SYBR green master mix (Invitrogen) was employed for quantitative reverse transcriptase PCR. *CgACT1* was used to normalize RNA levels and also served as endogenous control.

## Tandem mass tag-based quantitative total proteomics analysis

**Sample preparation**- For proteome analysis, YPD grown *C. glabrata* cells in exponential phase were pelleted at 3000 rpm, washed once with MiliQ, and resuspended in a lysis buffer containing 50 mM HEPES pH 8.2, 8 M Urea, 50 mM NaCl, and protease inhibitors (complete mini, EDTA-free). Cells were lysed using the Beadbeater in microfuge tubes for six cycles of 30 s each, with 1 min pauses between cycles to avoid overheating of the lysates. Lysates were cleared by centrifugation at 15000 rpm for 30 min at 4˚C. Protein concentration was then determined using the Lowry method and 200µg of proteins were run on a 10% SDS-PAGE to check the protein quality [72]. These proteins were then sent to Mass Spectrometry (MS) facilities, Institute of Bioinformatics, Bangalore, India. At the MS facilities, proteins were acetone precipitated, alkylated, and digested with trypsin.

For TMT labeling, the 12 channels from the TMT pro-16 plex kit (ThermoFisher Scientific) were used for labeling the respective samples. Each TMT channel reagent was dissolved in 200µl of anhydrous acetonitrile and incubated for 5 minutes. Then, 40 µl of respective TMT reagent was added to 80ug of each of the 12 sample peptide mix (dissolved in 40mM TEABC buffer), incubated for 60 mins at RT (after mixing) and the labeling reaction was stopped by adding 5% methylamine solution. An aliquot (5µl) of each of the labeled samples was quenched by adding 8 µl of 5% hydroxylamine, pooled, dried, reconstituted in 0.1% TFA, cleaned up using C18 stage tip, dried and re-dissolved in LCMS grade water (containing 0.1% TFA and 3% acetonitrile) for labeling efficiency check. After the C18 stage tip clean-up, the sample was dried and dissolved in water (containing 0.1% TFA and 3% acetonitrile) for LCMS analysis to check the labeling efficiency. The labeling efficiency (>95%) and reporter ion intensity normalization (median of TMT ratios for mixing equal amounts of TMT-labeled peptides) were performed. After the labelling efficiency check, reaction was quenched using 5% hydroxylamine (for 15 min, at RT). Based on the reporter ion intensity values, each of the labeled samples was pooled, dried, and fractionated using basic pH Reverse phase HPLC into 96 fractions and concatenated into 12 fractions. The final LCMS run was performed (in triplicates) after dissolving the (desalted) 12 fractions in water (containing 0.1% TFA and 3% acetonitrile) and loading 1µg equivalent peptide to the mass spectrometer [73,74].

The LCMS analysis was performed using Q Exactive HF-X (Thermo Scientific, Bremen, Germany) interfaced with Dionex Ultimate 3000 nanoflow liquid chromatography system. The acquired mass spectrometry data was analyzed against *C. glabrata* reference proteome

(downloaded from the UniProt database) using the SEQUEST search algorithm on the Proteome Discoverer platform (version 2.4, Thermo Scientific). Data was filtered with a 0.01% false discovery rate (FDR) based on the decoy search.

### Analysis of gene ontology

The corresponding *C. glabrata* systematic ORFs (CAGL IDs) were obtained from the UniProt Database (https://www.uniprot.org/) [75] using accession numbers of proteins that were determined by mass spectrometry analysis. Proteins were identified by using CAGL IDs from the Candida Genome Database (http://www.candidagenome.org/) [76]. Using the DAVID tools (https://david.ncifcrf.gov/tools.jsp) [77], functional enrichment analysis (biological processes, cellular components, and molecular functions) was carried out. Graphs were plotted by using GraphPad Prism and SRplot software.

### Microscopy

To study the mitochondrial morphology and reactive oxygen species (mtROS) in wild type and deletion strains of *C. glabrata*, MitoTracker Green dye (Invitrogen, M7514) and MitoSOX Red (Invitrogen, M36008) were used respectively. Also, we used N-acetylcysteine (NAC), a known antioxidant to check the reduction in ROS level. Cells grown in YPD medium (1.0 $A_{600}$) with or without 10mM NAC were harvested, and the pellet was washed thrice with 1x PBS to get rid of any remaining YPD medium [78]. After that, cells were treated with 200 nM MitoTracker Green and 1 μM MitoSOX Red in 250μl PBS and allowed to incubate for 20 minutes at 30˚C. Following incubation, cells were rinsed with PBS and resuspended in PBS without MitoTracker Green and MitoSOX Red. Cells resuspended in PBS were placed on a 35mm cover glass bottom dish that had been coated with 0.1% solution of concanavalin A to facilitate imaging. Images were captured using a Leica TCS SP8 confocal microscope equipped with an HC PL APO CS2 63x/1.40 oil objective. The microscopy data was processed and analyzed using Fiji Software.

### Statistical analysis

The data was analyzed using GraphPad Prism software, employing the student's t test and one-way analysis of variance (ANOVA). A *p* value of less than 0.05 was regarded as statistically significant.

### Supporting information

**S1 Fig. Schematic representation of orthologs of the *S. cerevisiae* STUbL proteins Slx5, Slx8, Uls1 in *Candida glabrata*.** Maps of proteins with their domains were produced using IBS 1.0.
(PDF)

**S2 Fig. Quantification of growth of STUbLs mutants.** The indicated strains were plated out on the specific treatment plates and the colonies that appeared were counted and normalized against counts obtained from untreated plates. Data represent the mean of biological triplicates and the error bars indicate standard deviation.
(TIF)

**S3 Fig. Complementation of STUbL mutants.** To test growth under various stress conditions, a serial dilution-spotting assay was done. 3 μl of 10 fold serial dilutions of cultures at $A_{600}$ 1 of the indicated *C. glabrata* strains were spotted on different media. Images of paltes were taken

after 2 days of incubation at 30˚C and 37˚C, 3 days of incubation with methylmethanesulfonate (MMS), and hydroxyurea (HU), at concentrations of 0.04%, 100mM, and at higher temperature (42˚C), respectively.
(TIF)

**S4 Fig. Growth of *C. glabrata* STUbL deletions in YPD and RPMI.** CFU to assess cell proliferation of the indicated *C. glabrata* strains in YPD (A, B) and RPMI-1640 medium containing 10% FBS (fetal bovine serum) (C, D). Yeast cells were collected at the indicated time points (2, 8, 12, and 24 hours), serially diluted, and then plated on YPD medium. Following 24–48 hours of incubation at 30˚C, yeast colonies were counted. Data shows the mean and SD of three independent experiments.
(TIF)

**S5 Fig. A. Complementation of *Cguls1Δ* mutant with plasmid encoding HF-CgUls1.** 3μl of 10 fold serial dilutions of cultures at $A_{600}$ 1 of the indicated *C. glabrata* strains were spotted on CAA plates. Plate images were taken after 2–3 days of incubation. **B.** CgUls1p is elevated in STUbL deletion mutants. The bar graph illustrates the $\log_2$ fold change of CgUls1 protein levels in the mutants compared to the wild type as obtained in the mass spectrometry. Error bars representing the $\log_2$ standard deviation.
(TIF)

**S6 Fig. Comparison of levels of proteins involved in various transport processes in STUbL deletion mutants.** Graph shows the change in levels of proteins involved in various transport pathways in the *Cgulp2Δ*, *Cgulp2ΔCgslx8Δ*, and *Cgulp2ΔCguls1Δ* mutants in comparison to wild type. The error bar represents the $\log_2$ standard deviation.
(TIF)

**S7 Fig. Heat map illustrating the differences in protein levels of ribosomal proteins.**
**A.** Heat map of mitochondrial ribosomal proteins and **B.** cytoplasmic ribosomal proteins of large and small subunits in *Cgulp2Δ*, *Cgslx8Δ*, *Cguls1Δ* single mutants, *Cgulp2ΔCgslx8Δ* and *Cgulp2ΔCguls1Δ* double mutants in comparison with WT. Heat maps were generated for the increased (shades of red color) and decreased (shades of green color) levels of proteins for each mutant with respect to wild type.
(TIF)

**S8 Fig. Heat map illustrating the altered levels of the components of protein degradation pathways in *Cgulp2Δ*, *Cgslx8Δ*, *Cguls1Δ* single mutants, *Cgulp2ΔCgslx8Δ* and *Cgulp2ΔCguls1Δ* double mutant in comparison with WT.** A heat map of the elevated (shades of red color) and diminished (shades of green color) levels of the components of protein degradation pathways including the proteasome and ERAD pathway for each mutant.
(TIF)

**S9 Fig. Comparison of levels of proteins involved in biosynthetic processes.** Heat map shows the changes in levels of proteins involved in biosynthetic processes in each mutant in comparison to wild type with increase indicated in red and decrease in green.
(TIF)

**S10 Fig. Comparison of levels of proteins involved in metabolic processes.** Heat map shows the changes in levels of proteins involved in metabolic processes in each mutant in comparison to wild type with increase indicated in red and decrease in green.
(TIF)

**S11 Fig. A.** Expression of FLAG tagged CgAde5,7. Western blot was performed to check the expression of CgAde5,7 protein in WT and mutants using α-flag antibody. The six lanes preceding the molecular weight marker are negative controls of WT and mutants with no tag and the lanes following the marker are strains carrying plasmid encoding FLAG tagged CgAde5,7. Tubulin is used as loading control. **B.** Effect of purine synthesis inhibitors on growth of *C. glabrata*. Doubling times were calculated based on absorbance $A_{600}$ for 48 hours measured at regular intervals and are presented as a bar graph. Error bar represents the standard deviation.
(TIF)

**S1 Table. Orthologs of the *S. cerevisiae* STUbL proteins Slx5, Slx8, Uls1 in *Candida glabrata*.** Protein sequences were collected from Saccharomyces Genome Database [37] and Blastp was used to determine *C. glabrata* orthologs. Domains were annotated using Pfam. The EMBOSS Stretcher (pairwise sequence alignment) tool was used to calculate the percentages of identity and similarity.
(TIF)

**S2 Table. A list of 107 significantly elevated proteins and 90 significantly reduced proteins identified in *Cgulp2Δ* mutant in comparison with wild type.**
(XLSX)

**S3 Table. A list of 87 significantly elevated proteins and 130 significantly reduced proteins identified in *Cgslx8Δ* mutant in comparison with wild type.**
(XLSX)

**S4 Table. A list of 148 significantly elevated proteins and 241 significantly reduced proteins identified in *Cguls1Δ* mutant in comparison with wild type.**
(XLSX)

**S5 Table. A list of 278 significantly elevated proteins and 184 significantly reduced proteins identified in *Cgulp2ΔCgslx8Δ* mutant in comparison with wild type.**
(XLSX)

**S6 Table. A list of 300 significantly elevated proteins and 134 significantly reduced proteins identified in *Cgulp2ΔCguls1Δ* mutant in comparison with wild type.**
(XLSX)

**S7 Table. Enriched GO terms for biological process (BP), cellular component (CC) and molecular function (MF) categories, as determined by the DAVID tool, for the elevated and reduced proteins in *Cgulp2Δ* mutant.**
(XLSX)

**S8 Table. Enriched GO terms for biological process (BP), cellular component (CC) and molecular function (MF) categories, as determined by the DAVID tool, for the elevated and reduced proteins in *Cgslx8Δ* mutant.**
(XLSX)

**S9 Table. Enriched GO terms for biological process (BP), cellular component (CC) and molecular function (MF) categories, as determined by the DAVID tool, for the elevated and reduced proteins in *Cguls1Δ* mutant.**
(XLSX)

**S10 Table. Enriched GO terms for biological process (BP), cellular component (CC) and molecular function (MF) categories, as determined by the DAVID tool, for the elevated**

**and reduced proteins in *Cgulp2ΔCgslx8Δ* mutant.**
(XLSX)

**S11 Table. Enriched GO terms for biological process (BP), cellular component (CC) and molecular function (MF) categories, as determined by the DAVID tool, for the upregulated and downregulated proteins in *Cgulp2ΔCguls1Δ* mutant.**
(XLSX)

## Acknowledgments

The authors thank Dr. Rupinder Kaur, CDFD, for plasmids and strains. We thank Institute of Bioinformatics, Bangalore for quantitative proteomics. Authors acknowledge infrastructure support from DST-FIST SR/FST/LS-11/2023/1172C and DBT-SAHAJ-Builder (BT/INF/22/SP41176/2020).

## Author Contributions

**Conceptualization:** Krishnaveni Mishra.

**Data curation:** Dipika Gupta.

**Formal analysis:** Dipika Gupta, Krishnaveni Mishra.

**Funding acquisition:** Krishnaveni Mishra.

**Investigation:** Dipika Gupta.

**Methodology:** Dipika Gupta, Renu Shukla.

**Project administration:** Krishnaveni Mishra.

**Resources:** Krishnaveni Mishra.

**Supervision:** Krishnaveni Mishra.

**Validation:** Dipika Gupta, Renu Shukla.

**Writing – original draft:** Dipika Gupta, Krishnaveni Mishra.

**Writing – review & editing:** Dipika Gupta, Krishnaveni Mishra.

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
