## [Decision Letter · Decision Letter 0]

5 Sep 2024

Dear Dr. Mishra,

Thank you very much for submitting your manuscript "SUMO-targeted Ubiquitin Ligases regulate protein homeostasis and pathogenesis of Candida glabrata" for consideration at PLOS Pathogens. As with all papers reviewed by the journal, your manuscript was reviewed by members of the editorial board and by several independent reviewers. In light of the reviews (below this email), we would like to invite the resubmission of a significantly-revised version that takes into account the reviewers' many insightful comments, which are clearly explained and do not need comment from my side. Two reviewers also suggest to test the virulence of the mutants in animal infection models, since the paper claims that the STUbLs regulate pathogenicity. I would not be surprised if the disturbance of a central cellular function reduces the fitness of a pathogen in its host and do not insist on such experiments. If you do not wish to add animal infection experiments, however, the title of the paper and corresponding statements in the text should be modified accordingly.

A question that occurred to me when reading the paper: Do you have an explanation for why a proteasome inhibitor (MGI32) does so strongly (ca. 10-fold increase in CFUs) improve the growth of C. glabrata wild-type cells even in the absence of stress conditions (Fig. 6E)?

We cannot make any decision about publication until we have seen the revised manuscript and your response to the reviewers' comments. Your revised manuscript is also likely to be sent to reviewers for further evaluation.

Sincerely,

Joachim Morschhäuser

Academic Editor

PLOS Pathogens

Alex Andrianopoulos

Section Editor

PLOS Pathogens

Michael Malim

Editor-in-Chief

PLOS Pathogens

orcid.org/0000-0002-7699-2064

Reviewer's Responses to Questions

**Part I - Summary**

Reviewer #1: In this study, the authors investigated the roles of SUMOylation in Candida glabrata. They constructed deletion mutants of various genes involved in this pathway and conducted phenotypic characterizations. They found that Ulp2, a deSUMOylase, is essential for maintaining protein homeostasis, and that the loss of both Ulp1 and the ubiquitin ligase Slx8 leads to heightened protein degradation, making the cells susceptible to various stressors, such as high temperature and DNA-damaging agents. Importantly, ulp2, slx6, and ulp2 slx8 mutants fail to proliferate in macrophages. Proteomic analyses revealed that these mutations significantly compromise purine biosynthesis and mitochondrial functions.

This is the first comprehensive study of the role of SUMOylation in C. glabrata. It identified genes and cellular processes crucial for stress tolerance and survival in macrophages, suggesting potential targets for antifungal therapies.

While the data suggest that genes such as ULP2 and SLX8 may also be important for pathogenicity, the study does not include data on virulence using animal infection models.

To enhance the readability of the paper, the authors should improve their use of proper punctuation, which is missing in many places.

Reviewer #2: The manuscript by Gupta et al presents findings that characterise the roles of the deSUMOylase Ulp1 and the SUMO-targeting ubiquitin ligases Slx5, Slx8 and Uls1 in Candida glabrata. Characterisation of a set of single and double mutants in genes encoding these proteins is presented (e.g. growth under various conditions, stress responses, growth within immune phagocytes), alongside biochemical studies and proteome analyses to understand the growth phenotypes of the mutants. The study presents a relatively detailed analyses of the phenotypes of the mutants in vitro, and the biochemistry supports the conclusion that these proteins, as expected, are required for protein homeostasis. As a consequence, inactivation of these pathways leads to pleiotropic phenotypes. Mitochondrial morphology and function, as well as purine biosynthesis seems to be particularly affected. At the level of mechanism, the data is consistent with Ulp1 and not Slx5/Slx8 being the main SUMO-targeting ubiquitin ligase for proteins that are de-ubiquitinated by Ulp1. In contrast enhanced degradation of Ulp1 targets through the proteasome pathways is seen in the ulp1 slx8 mutant, potentially explaining the various phenotypes.

Reviewer #3: In this report, Gupta et al. follow up on their analysis of the C. glabrata desumoylase Ulp2, and show interesting genetic interactions between the disturbance of protein sumoylation caused by deletion of ULP2, and the function of the SUMO-dependent ubiquitin ligases (STUbL) Uls1 and Slx5/Slx8. In particular, they find that the ulp2 slow growth phenotype can be suppressed by further deletion of ULS1 (but not of SLX8), under several conditions (although under other conditions, such as hydroxyurea, the phenotype is exacerbated). They go on to show that the level of SUMOylated proteins, which is reduced in ulp2, is recovered in the ulp2 uls1 mutant, leading to the suggestion that the phenotype of the ulp2 deSUMOylase mutant is due to increased protein degradation vie the STUbL Uls1 and the proteasome. This model is then corroborated by showing that addition of the proteasome inhibitor MG132 can rescue both SUMOylated protein levels and growth in the ulp2 mutant. The authors then carried out mass spec analysis of the different single and double mutants and focus on a few pathways expected, based on the MS data, to be affected: purine biosynthesis and mitochondrial function. Additional observations are that all the mutants analyzed become sensitive to killing by macrophages and are defective in biofilm formation.

This paper contains many interesting observations, but I found it to be a very difficult read. One problem is that the authors spend a lot of time framing their work as addressing the antifungal drug crisis in general, and in the Candida clade (they emphasize C. auris) in particular. In reality however, C. glabrata is phylogenetically much closer to S. cerevisiae than to all other Candida spp., so that generalization of any result from C. glabrata to other Candida spp. is problematic. And while one might argue that C. glabrata is an important pathogen in its own right and devising therapies against it is a worthwhile goal, the notion that a universally conserved pathway (the human host also contains deSUMOylases and STUbLs) represents a useful drug target is not convincing. A second problem is that while the paper contains a coherent and interesting core, namely the observations described in Figs. 2, 4, 5 and 6, as well as the MS analysis, I find the follow up on the MS analysis (Figs. 8 and 9), disconnected from the main story and, in the case of purine biosynthesis, unconvincing (see detailed comments below). I would recommend an extensive re-write that would put the focus on the molecular biology aspects of the SUMO/STUbL/proteasome axis.

**Part II – Major Issues: Key Experiments Required for Acceptance**

Reviewer #1: (1) In the ulp2 slx8 mutant, mitochondrial complexes II, III, IV, and V were significantly downregulated. Since mutations affecting the mitochondrial ETC have been linked to antifungal resistance in various Candida species, including C. glabrata (e.g., 10.1111/mmi.15229), the authors should also perform antifungal susceptibility tests on the ulp2 and slx8 ulp2 mutants, given their resistance to various stresses.

(2) Despite the demonstrated importance of ULP2 and SLX8 in stress tolerance and survival macrophages, this study provides limited insights into the pathogenicity of C. glabrata. It would be helpful to provide some data on virulence using animal infection models.

Reviewer #2: My comments and suggestions are below.

• I might have missed this, but from a mechanism point of view, since Slx8 is a ubiquitin ligase it is unclear why protein degradation would be enhanced in its absence. It seems as if the suggestion is that upregulation of Uls1 is the possible reason. However, this is only seen in the ulp2 slx8 mutant, and therefore might be indirect. Further discussion on this point would be beneficial.

• Given the relatedness of C. glabrata and S. cerevisiae, and the high conservation of the pathways under study across organisms, I would suggest that it would be beneficial to present a more detailed discussion on the conserved vs diverged roles of Ulp2, Slx5 /Slx8 and Uls1 in C. glabrata compared to what we know from other organisms in terms of mechanism, mutant phenotypes, genetic interactions, essentiality of genes or not etc.

• Changes in mitochondrial morphology presented in Figure 9 are not obvious. I would suggest more detailed microscopy images of representative cells and quantification of the changes.

• Please explain in figure legends the nature of the replicates (biological or technical) and define what the error bars represent. At present, this information is available for some panels, but not all (across all figures).

Reviewer #3: 1) Fig. 1 and Table 1 – in silico protein comparison – is not very informative. I suggest displaying it as supplementary information, if at all.

2) Fig. 4 and p. 13: here and throughout the paper, the authors refer to the SUMO-reactive bands on the Western blot as “poly-SUMOylation”. However it is not clear to me why these bands couldn’t equally represent mono-SUMOylated proteins.

3) pp. 21-23 and Fig. 8: the authors found by MS that anabolic proteins, and in particular purine biosynthesis, are affected in the STUbL mutants. The protein ADE5,7 seems the most affected. However, is only reduced 10-20% in ulp2 and ulp2 slx8, and is up 10% in ulp2 uls1. To demonstrate that these difference are physiologically significant, the authors then proceed to show that overexpression of ADE5,7 can suppress the slow growth of ulp2 and ulp2 slx8, and that the STUbL mutants are more sensitive to purine pathway inhibitors. However, 1) while these mutant strains indeed exhibit better growth with the ADE5,7 overexpression plasmid, so does at 42° also the ulp2 uls1 strain, which has a higher level of the protein to begin with (Fig. 8C). In fact, according to Figs 8D, E, so do all the strains, including the wild type, regardless of ADE5,7 protein levels, suggesting this could be a plasmid effect. I also note that the ADE5/7 OE strains and their controls are not spelled out in either the legend or the Methods section. 2) In Fig. 8F-I, the authors show inhibition of growth by purine pathway inhibitors, and they suggest that the higher inhibition detected for some of the strains correlates with lower gene expression. However, it is hard to gauge, from the figures shown, which differences are significant. Furthermore, contrary to expectation, no resistance is detected in the naturally ADE5,7 overexpressing strain ulp2 uls1. Thus, I do not find that the authors have convincingly show physiological relevance of the MS data regarding the purine biosynthesis pathway.

4) Looking at the proteasome proteins just in ulp2 slx8 (Fig. S7) vs the Excel sheet S4, I see that Pup1 is most induced in the Excel, but appears black in Fig. S7; while the bright red Hlj1 is only moderately induced according to the Excel. I recommend to re-check all S6-S9 figures vs. the data in the Excel sheet.

**Part III – Minor Issues: Editorial and Data Presentation Modifications**

Reviewer #1: Line 116 at least

Lines 118-119. To understand the importance of STUbLs in C. glabrata,,

Ulp2 was described as SUMO specific isopeptidases, deSUMOylase, deSUMOylating enzyme, or SUMO protease in different places, which could be confusing for readers who are not familiar with topic.

Line 143 and throughout the manuscript, the prefix 'Cg' can be removed from all gene and protein names, as readers are unlikely to confuse them with those from other species. The prefix should be used only when two or more species are being compared or described together. This will make the text easier to read and the figures less crowded.

Figure 2 The number of strains shown in the heat map in Figure 2D exceed that used in panels A-C. The ‘+’ symbol used in the colour scale does not make sense and should be removed together with the ‘–’ symbol.

Line 239. The description is not clear. After 8 hours of what? No matter what, it is not reflected in Figure 3A and B.

Figure 3A-B. I suggest changing ‘Macrophage internalized cells’ to ‘Survival in macrophage’.

Line 261. Replace the asterisk symbol with X. Also, change ‘C. glabrata strains to ‘C. glabrata cells’.

Line 266. in macrophage-internalized

Line 284. at least

Line 297. Ponceau S

Line 447. The citation format here is different from the rest.

Figure 8A. The bars of different colours, red, green and blue, are not explained.

Lines 498-499. How the cells were grown was not described. In liquid medium? how long was the growth period?

Figure 8F-H. The Y-axis represents the percentage of inhibition. What does the height of the grey bars indicate? Typically, the growth of the vehicle control should be set to one, and the growth of the drug-treated cells should be expressed as a percentage relative to the vehicle control.

Line 513. mitochondrial not moitochondrial.

Lines 513-516, This long sentence needs proper punctuation marks. Line 514, in the Cgulp2ΔCgslx8Δ double mutant as complexes II, III, IV, and V.

Line 612. a few

Line 653. homologues

Line 668. uncover new targets for antifungal therapies.

Line 669. Replace ‘a lot of’ with ‘multiple’, and ‘at once’ with ‘simultaneously, ’.

Lines 671-672. …homeostasis, offering an additional approach for designing combinatorial therapies.

Lines 672-673. Please provide the references for the statement ‘For instance, some of the purine biosynthesis inhibitors are already used as antifungals in the clinic’.

Line 676. design of

Reviewer #2: (No Response)

Reviewer #3: 1) Slx5 and Slx8 are expected to function as a heterodimer , and indeed this is how the authors analyze them. However the phenotype of the individual slx5 and slx8 mutants are not always coherent, most notably at 42° and at the level of SUMOylation (Fig. 4A). This merits discussion.

2) I couldn’t help noticing, while scanning the MS data, that Smt3/SUMO itself is significantly reduced (about 35%) in all mutants except uls1. I wonder how this observation can be fitted in the author’s model of the SUMO/STUbL/proteasome axis.

3) The observation in Fig. S7 and Fig. 7 that the proteasome proteins are over-represented in ulp2 uls1 and under-represented in ulp2 slx8 correlates nicely with the absence and presence of sumoylated bands in the Westerns of these strains, and with the MG132 effect. I didn’t see this observation discussed in the text.

4) The color scheme for “up” and “down” proteins is opposite in Fig. 7 vs. Fig. S6-S9. This is confusing.

5) For Fig. 7, the significance of the X axes is not clear. Does "% protein" mean average protein levels or number of proteins in the category that are up or down? If the latter, then putting the percentages in the negative range of the axis doesn't make sense.

6) Since the paper deals only with C. glabrata, the Cg prefix is not really required and removing it from all gene and protein names might make the text more readable.

Typos

l. 24: opportunistic

l. 71: should be “the epsilon amino group of lysine in target proteins”

l. 81: isopeptidases

l. 116: at least

l. 169: should be “The Cgulp2ΔCgslx8Δ strain was …”

l. 513 moitochondrial

l. 567 Molecualr

throughout: “thermal sensitivity” should be “temperature sensitivity”

PLOS authors have the option to publish the peer review history of their article (what does this mean?). If published, this will include your full peer review and any attached files.

Reviewer #1: No

Reviewer #2: No

Reviewer #3: No
---

## [Decision Letter · Decision Letter 1]

11 Nov 2024

Dear Dr. Mishra,

We are pleased to inform you that your manuscript 'SUMO-targeted Ubiquitin Ligases as crucial mediators of protein homeostasis in Candida glabrata' has been provisionally accepted for publication in PLOS Pathogens.

Before your manuscript can be formally accepted you will need to complete some formatting changes, which you will receive in a follow up email. A member of our team will be in touch with a set of requests. Please take this opportunity also to correct the typing errors pointed out by reviewer 3.

Best regards,

Joachim Morschhäuser

Academic Editor

PLOS Pathogens

Alex Andrianopoulos

Section Editor

PLOS Pathogens

Michael Malim

Editor-in-Chief

PLOS Pathogens

orcid.org/0000-0002-7699-2064

Reviewer Comments (if any, and for reference):

Reviewer's Responses to Questions

**Part I - Summary**

Reviewer #1: This revised manuscript reflects the authors' efforts to address all the issues I previously raised, resulting in a significantly improved version. I appreciate their work in making these revisions.

Reviewer #2: The authors have addressed my comments. I have no further suggestions.

Reviewer #3: In this revision the authors corrected the errors and modified the text to partially de-emphasize the antifungal development angle. The paper is still lacking in coherence: the proposed interactions within the SUMO-STUbL-proteasome axis are interesting, but they are still not fleshed out mechanistically, and the mitochondrial and purine biosynthesis phenotypes of the mutants are described but not explained, resulting in a very descriptive report. This is reflected in the proposed model, which is quite vague.

**Part II – Major Issues: Key Experiments Required for Acceptance**

Reviewer #1: N.A.

Reviewer #2: (No Response)

Reviewer #3: (No Response)

**Part III – Minor Issues: Editorial and Data Presentation Modifications**

Reviewer #1: N.A.

Reviewer #2: (No Response)

Reviewer #3: Typos:

l. 240: macrophages

l. 545 magenta

l. 611 “compared to the Cgulp2Δ mutant”

PLOS authors have the option to publish the peer review history of their article (what does this mean?). If published, this will include your full peer review and any attached files.

Reviewer #1: No

Reviewer #2: No

Reviewer #3: No

---

## [Editor Report · Acceptance letter]

28 Nov 2024

Dear Dr. Mishra,

We are delighted to inform you that your manuscript, "SUMO-targeted Ubiquitin Ligases as crucial mediators of protein homeostasis in Candida glabrata," has been formally accepted for publication in PLOS Pathogens.

Best regards,

Michael Malim

Editor-in-Chief

PLOS Pathogens

orcid.org/0000-0002-7699-2064